# Identifying latent distances with Finslerian geometry

## Abstract

Riemannian geometry provides powerful tools to explore the latent space of generative models while preserving the inherent structure of the data. Distance and volume measures can be computed from a Riemannian metric defined by pulling back the Euclidean metric from the data to the latent manifold. With this in mind, most generative models are stochastic, and so is the pullback metric. Yet, manipulating stochastic objects is at best impractical, and at worst unachievable. To perform operations such as interpolations, or measuring the distance between data points, we need a deterministic approximation of the pullback metric. In this work, we define a new metric as the expected length derived from the stochastic pullback metric. We show this metric defines a Finsler metric. We compare it with the expected pullback metric. We show that in high dimensions, the metrics converge to each other at a rate of $\mathcal{O}\left(\frac{1}{D}\right)$.

## 1 Introduction

Generative models provide a convenient way to learn low-dimensional latent variables $z$ corresponding to data observations $x$ through a smooth function $f : \mathcal{Z} \subset \mathbb{R}^q \to \mathcal{X} \subset \mathbb{R}^D$, such that $x = f(z)$. In practice, this function can be a Gaussian Process Latent Variable Model (GPLVM) (Lawrence, 2003) or similar.

Through this learnt manifold, one can generate new data or compare observations by interpolating or computing distances. However, doing so by using the Euclidean distance in the latent space is misleading (Hauberg, 2018a). If our observations are lying near a manifold (Fefferman et al., 2016), we want to equip our latent space with a metric that preserves distance measures on it. Figure 1 (left panel) illustrates the need for defining geometric-aware distances on manifolds.

Distances on a manifold can be precisely defined using a norm, which is a mathematical function that exhibits several desirable properties such as non-negativity, homogeneity, and the triangle inequality. In particular, a norm can be induced by an inner product (i.e., a quadratic function) that associates each pair of points on the manifold with a scalar value.

Let us compute the infinitesimal Euclidean norm in our data space. Using the Taylor expansion, we have: $\left\|f(z + \Delta z) - f(z)\right\|_2^2 \approx \left\|f(z) + J(z)\Delta z - f(z)\right\|_2^2 = \Delta z^\top J(z)^\top J(z)\Delta z$. As a first approximation, the norm defined in the latent space locally preserves the Euclidean norm defined in the data space. The curvature of our data manifold is condensed in the Riemannian metric tensor $G_z = J^\top(z)J(z)$, which serves as a proxy to define the Riemannian metric: $g_z : (u, v) \to u^\top G_z v$. In mathematical jargon, we say that the Riemannian manifold $(\mathcal{Z}, g)$ is obtained by pulling back the Euclidean metric through the map $f$.

Riemannian geometry enables the exploration of the latent space in precise geometric terms, and quantities of interest such as the length, the energy or the volume can be directly derived from the pullback metric. These geometric quantities are, by construction, known to be invariant to reparametrizations of the latent space $\mathcal{Z}$, and are thus statistically identifiable. Yet, we encounter another problem: while this geometric framework exclusively handles deterministic objects, the decoding part of generative models is often stochastic. The learnt map $f$, that mathematically describes those decoders, is stochastic too, and so is the Riemannian metric pulled back through it. This is shown in the right panel of Figure 1. In conclusion, in order to navigate our latent manifold, **we need a deterministic approximation of our pullback metric.**

Previous research has approximated the stochastic pullback metric with the expected value of the Riemannian metric tensor. Yet, the metric tensor serves as a surrogate quantity to define an induced norm on our manifold. Instead of taking the expectation of the metric tensor, we propose to take the expectation of the norm directly. **In this paper**, we compare our expected norm with the norm induced by the expected metric tensor. The main findings are:

1. The expected norm defines a Finsler metric. Finsler geometry is a generalisation of Riemannian geometry.

2. For Gaussian Processes, the stochastic norm obtained through the pullback metric follows a non-central Nakagami distribution, so our Finsler metric has a closed-form expression.

3. In high dimensions, for Gaussian Processes, our Finsler metric and a previously studied Riemannian metric converge to each other at a rate of $\mathcal{O}\left(\frac{1}{D}\right)$.

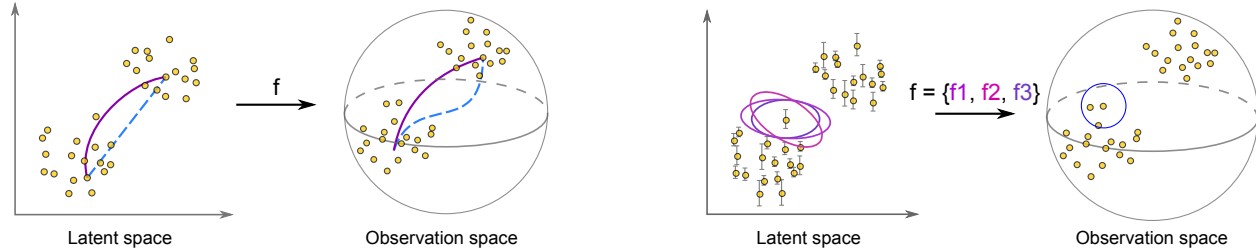

Figure 1: **Left figure**: The Euclidean distance measure, in blue, in the latent space does not take into consideration the geometry of the observational manifold, and therefore it is not identifiable and leads to misinterpretations. Instead, the length derived from the pullback metric will follow the curvature of the manifold. **Right figure**: Generative models often map the latent space to the data space using a stochastic process $f = \{f_1, f_2, \dots\}$. A stochastic Riemannian metric, whose realisations are represented by ellipses, are obtained when we pull back the Euclidean metric, represented by a unit circle, through the stochastic process $f$.

## 1.1 Outline of the paper

The paper explores the latent spaces learned by generative models, which encode a latent low-dimensional manifold that represents observed high-dimensional data. The latent manifold is noted $\mathcal{Z} \subset \mathbb{R}^q$ and the data manifold is noted $\mathcal{X} \subset \mathbb{R}^D$.

Assuming the manifold hypothesis holds, we need to define a norm in the latent manifold to compute distances that respect the underlying geometry of the data. Such a norm can be constructed by pulling back the Euclidean distance through the smooth function that maps the latent manifold to the data manifold. This map, $f : \mathcal{Z} \to \mathcal{X}$, mathematically describes the decoder of a trained generative model. Those models being often stochastic, we consider $f$ being a stochastic process. It means that the pullback metric tensor, $G = J^\top J$, and its induced norm, $\|\cdot\|_G : u \to \sqrt{u^\top G u}$, are also stochastic. In section 2, we mathematically define the notion of stochastic pullback metric and stochastic manifolds.

To circumvent all the challenges posed by this stochastic component, a deterministic approximation of the norm is needed. It can be defined by taking the expectation of the metric tensor. This norm, that we will note $\|\cdot\|_R : u \to \|u\|_{\mathbb{E}[G]}$, has been studied before by Tosi et al. (2014); Arvanitidis et al. (2018), and is explained in section 2.2. In this paper, we propose instead to directly take the expectation of the stochastic norm. This expected norm, noted $\|\cdot\|_F : u \to \mathbb{E}\left[\|u\|_G\right]$, is introduced in section 2.4. The norm $\|\cdot\|_R$ is defined by a Riemannian metric, and we show that the norm $\|\cdot\|_F$ defines a Finsler metric. We explain the general difference between Finsler and Riemannian geometry in section 3.

The aim of this paper is to compare those two norms. We first draw absolute bounds in section 3.2, and then relative bounds in 3.3. We also investigate the relative difference of the norms when the dimension of the data space inscrease, in section 3.4. Finally, we perform some experiments in Section 4, that illustrates the Riemannian and Finsler norm in the same latent space.

## 1.2 Related works

**Finsler geometry in Machine Learning**

Our work crucially relies on Finslerian geometry, which has been well-studied mathematically, but has only seen very limited use in machine learning and statistics. We point to two notable exceptions, which are quite distinct from our work. Lopez et al. (2021) use symmetric spaces to represent graphs and endow these with a Finsler metric to capture dissimilarity structure in the observational data. Ratliff et al. (2021) discuss the role of differential geometry in motion planning for robotics. Along the way, they touch upon Finslerian geometry, but mostly as a neat tool to allow for generalizations. To the best of our knowledge, no prior work has investigated the links between stochastic and Finslerian geometry.

**Strategies to deal with stochastic Riemannian geometry**

Tosi et al. (2014) and Arvanitidis et al. (2018) introduced approximation of the pullback metric by taking the expectation of the metric tensor. In those two cases, the map $f$ is respectively a trained Gaussian process, or the decoder of a VAE. In this paper, the derivations only hold if $f$ is a smooth stochastic process (Definition 2.2), which is not the case of the VAEs[1], and hence, our results cannot be applied for those models.

In addition to the work of Tosi et al. (2014), a solution to circumvent the randomness of the metric tensor is to consider that the data follows a specific probability distribution. Instead of looking at the shortest path on the data manifold, Arvanitidis et al. (2021) borrow tools from information geometry and consider the straightest paths on the manifold whose elements are probability distributions.

## 2 Expectation on random manifolds

The metric pulled back by a stochastic mapping is, de facto, stochastic and endows a random manifold. Unfortunately, we are not yet equipped to derive geometric objects on a random manifold. Instead, we dodge this problem by seeking a deterministic approximation of this stochastic metric.

As mentioned above, a common solution is to approximate such a metric by its expectation. In section 2.2, we study the expected Riemannian metric and summarise the main findings of Eklund & Hauberg (2019).

The other solution suggested by this paper is to approximate the expectation of the lengths instead of the random metric itself. In section 2.4, we show that this new metric is not Riemannian but Finslerian (Proposition 2.2), and it has a closed-form expression when the map $f$ is a Gaussian process (Proposition 2.3).

### 2.1 Random Riemannian geometry

The pullback metric is appropriately defined as an Riemannian metric if and only if the mapping $f$ is an immersion, which is a differentiable function whose derivatives are injective everywhere on the manifold (Lee, 2013, Proposition 13.9). A manifold equipped with a Riemannian metric is called a Riemannian manifold.

**Definition 2.1.** The pullback of the Euclidean metric through the immersion $f : \mathcal{Z} \to \mathcal{X}$ is a **Riemannian metric**. It is defined as the inner product $g_z : (\mathcal{T}_z\mathcal{Z}, \mathcal{T}_z\mathcal{Z}) \to \mathbb{R}_+ : (u, v) \to u^\top G v$, at a specific point $z$ in the manifold $\mathcal{Z}$. $u$ and $v$ are vectors lying in the tangent plane $\mathcal{T}_z\mathcal{Z}$ (ie: the set of all tangent vectors) of the manifold. $G = J^\top J$, with $J$ the Jacobian of $f$.

---

[1]the decoder of a VAE, while it decodes to a Gaussian, cannot be considered as a differentiable stochastic process. One reason is because the independence of the probability of the data: $p(x|z) = \prod_{i=1}^{n} p(x_i|z_i)$. Let us assume the opposite: the decoder is a Gaussian process. The covariance of the Gaussian process would be a diagonal matrix because of the independence of the probability of the data. The covariance would correspond to a dirac distribution: $\text{cov}(x_i, x_j) = \delta_{ij}$. However, a stochastic process is differentiable only if the covariance is differentiable, which is not the case of the dirac distribution. Hence the decoder of a VAE cannot be considered as a differentiable stochastic process.

Since a Riemannian metric is an inner product, it induces a norm, noted $\|\cdot\|_G$. We can use this norm to define the **curve length** and **curve energy** on a manifold: $L_G(\gamma) = \int_0^1 \|\dot{\gamma}(t)\|_G \, dt$ and $E_G(\gamma) = \int_0^1 \|\dot{\gamma}(t)\|_G^2 \, dt$, with $\gamma$ a curve defined on $\mathcal{Z}$, and $\dot{\gamma}$ its derivative. A locally length-minimising curve between two connecting points is called a **geodesic**. To obtain a geodesic, we can minimise the curve length, but in practice minimising the curve energy is more efficient. On the manifold, we also may want to integrate probability functions, and so we need to define a **volume measure** that can be used akin to the change of variable formula for integrals: for $\mathcal{U} \subset \mathcal{Z}$, $\int_{f(\mathcal{U})} h(x) \, dx = \int_{\mathcal{U}} h(f(z)) V_R \, dz$, with $V_R(z) = \sqrt{G_z}$ the volume measure.

In addition, we are considering the case where the immersion $f$ is a stochastic process. The outputs of our trained model, $x \in \mathcal{X}$, which represent our data, are random variables.

> **Definition 2.2.** A **stochastic process** is a collection of random variables $\{X(t, \omega), t \in T\}$ indexed by an index set $T$ defined on a sample space $\Omega$, which represents the set of all possible outcomes. An outcome in $\Omega$ is denoted by $\omega$, and a realisation of the stochastic process is the sequence of $X(\cdot, \omega)$ that depends on the outcome $\omega$.

In this framework, our index set is our latent manifold $T = \mathcal{Z}$, and our sample space $\Omega$ is defined as the set of the model evaluations. For every point $z \in \mathcal{Z}$, every time we execute our model, the output $x = f(z)$ is a random variable following a specific distribution. When the data $x$ follow a Gaussian distribution, the stochastic process is called a **Gaussian process**. A GP-LVM is a model that learns how to map the data from a latent space to a data space through a Gaussian process.

When $f$ is a stochastic immersion, the metric tensor becomes a random matrix. In this paper, we call a manifold equipped with the stochastic pullback metric a **random manifold**, noted $(\mathcal{Z}, g)$. As a consequence of the stochastic aspect of the metric, all the functionals are stochastic themselves, and they are no longer trivial to manipulate.

> **Definition 2.3.** A **random Riemannian metric tensor** is a matrix-valued random field (ie: a collection of matrix-valued random variables $\{G(z, \omega), z \in \mathcal{Z}\}$), whose realisation for a specific evaluation $\omega \in \Omega$ is a Riemannian metric tensor. A **random Riemannian metric** is a metric induced by a random Riemannian metric tensor: $g_z : (\mathcal{T}_z\mathcal{Z}, \mathcal{T}_z\mathcal{Z}) \to \mathbb{R}_+ : (u, v) \to u^\top G v$. For the rest of the paper, the associated **stochastic norm** is noted:
>
> $$\|\cdot\|_G : \mathcal{T}_z\mathcal{Z} \to \mathbb{R}_+ : u \to \sqrt{g_z(u, u)} := \sqrt{u^\top G u}$$

If this stochastic norm is induced by $f$ defined as a Gaussian process, then $\|\cdot\|_G$ follows a non-central Nakagami distribution. This is explained in the proof of Proposition 2.3.

## 2.2 Norm induced by the expected metric tensor

One way to approximate a random metric tensor is to take its expectation with respect to the collection of random metrics induced by the stochastic process. This has been introduced before by Tosi et al. (2014) GP-LVMs.

> **Definition 2.4.** Let $G$ be a stochastic Riemannian metric tensor on the manifold $\mathcal{Z}$. We refer to $\mathbb{E}[G]$ as the **expected metric tensor**. It induces a Riemannian metric and a norm on $\mathcal{Z}$. We will note the **norm induced by the expected metric tensor** as:
>
> $$\|\cdot\|_R : \mathcal{T}_z\mathcal{Z} \to \mathbb{R}_+ : u \to \|u\|_{\mathbb{E}[G]} := \sqrt{u^\top \mathbb{E}[G] u}$$

Like any Riemannian metric, we can define the following functionals: $L_R(\gamma) = \int_0^1 \sqrt{\dot{\gamma}(t)^\top \mathbb{E}[G] \dot{\gamma}(t)} \, dt$, $E_R(\gamma) = \int_0^1 \dot{\gamma}(t)^\top \mathbb{E}[G] \dot{\gamma}(t) \, dt = \mathbb{E}[E_R(\gamma)]$, and $V_R(z) = \sqrt{\det \mathbb{E}[G]}$.

### 2.3 Expected paths on random manifolds

Approximating the stochastic metric by its expectation seems a natural but also ad-hoc solution. If we want to explore a manifold, we might prefer to use a representative quantity, such as the lengths between data points. This is the motivation of the work led by Eklund & Hauberg (2019). The expectation of the lengths can give us an idea about how, on average, two points are connected on a random manifold. The **expected curve length**, and its corresponding **curve energy** on the random manifold $(Z, g)$ are defined as: $L_F(\gamma) = \int_0^1 \mathbb{E}\left[\sqrt{\dot{\gamma}(t)^\top G \dot{\gamma}(t)}\right] dt = \mathbb{E}[L_R(\gamma)]$, and $E_F(\gamma) = \int_0^1 \mathbb{E}\left[\sqrt{\dot{\gamma}(t)^\top G \dot{\gamma}(t)}\right]^2 dt$.

One observation made by Eklund & Hauberg (2019) is that the length $(L_R)$ derived from the expected Riemannian metric is not equal to the expected curve length $(L_F)$, and their respective energy curves differ by a variance term:

$$
\begin{aligned}
E_R(\gamma) - E_F(\gamma) &= \int_0^1 \dot{\gamma}(t)^\top \mathbb{E}[G]\dot{\gamma}(t) - \mathbb{E}\left[\sqrt{\dot{\gamma}(t)^\top G \dot{\gamma}(t)}\right]^2 dt \\
&= \int_0^1 \mathbb{E}\left[\|\dot{\gamma}(t)\|_G^2\right] - \mathbb{E}\left[\|\dot{\gamma}(t)\|_G\right]^2 dt = \int_0^1 \mathrm{Var}\left[\|\dot{\gamma}(t)\|_G\right] dt
\end{aligned}
$$

This term can be regarded as a **regularisation term** for the Riemannian energy curve: the curve energy $E_R$ might be penalised when the curve goes through regions with high-variance. In practice, for a Gaussian process with a stationary kernel, this variance term is upper bounded by the posterior variance that is relatively low next to the training points and is high outside of the support of the data. Later, we will also see that the functionals agree in high dimensions, leading to the same geodesics (Section 3).

Eklund & Hauberg (2019) also noted that these quantities are bounded by the number of dimensions:

**Proposition 2.1.** (Eklund & Hauberg, 2019) Let $f : \mathbb{R}^q \to \mathbb{R}^D$ be a stochastic process such that the sequence: $\{f'_1, f'_2, \dots, f'_D\}$ has uniformly bounded moments. There is then a constant $C$ such that:

$$
0 \leq \frac{L_R - L_F}{L_R} \leq \frac{C}{8D}
$$

### 2.4 Expected norm and Finsler geometry

Our work can be seen as an extension of Eklund & Hauberg (2019)'s research. We are interested in approximating the stochastic norm instead of the metric tensor, and by doing so, the derived curve length and curve energy are the same ones studied by Eklund & Hauberg (2019). We go further as we not only compare curve lengths, but the deterministic norms obtained with the stochastic metric.

**Definition 2.5.** Let $G$ be a stochastic Riemannian metric tensor on the manifold $\mathcal{Z}$. It induces a stochastic norm, $\|\cdot\|_G$ on $\mathcal{Z}$. We will note the **expected norm** as:

$$
\|\cdot\|_F : \mathcal{T}_z\mathcal{Z} \to \mathbb{R}_+ : u \to \mathbb{E}[\|u\|_G] := \mathbb{E}\left[\sqrt{u^\top G u}\right]
$$

While it cannot be induced by an inner-product, it is sufficiently convex to be defined as a **Finsler metric**.

**Definition 2.6.** Let $F : \mathcal{T}\mathcal{Z} \to \mathbb{R}_+$ be a continuous non-negative function defined on the tangent bundle $\mathcal{T}\mathcal{Z}$ of a differentiable manifold $\mathcal{Z}$.

We say that $F$ is a **Finsler metric** if, for each point $z$ of $\mathcal{Z}$ and $v$ on $\mathcal{T}_z\mathcal{Z}$, we have (1) **Positive homogeneity**: $\forall \lambda \in \mathbb{R}_+$, $F(\lambda v) = \lambda F(v)$. (2) **Smoothness**: $F$ is a $C^\infty$ function on the slit tangent bundle $\mathcal{T}\mathcal{Z} \setminus \{0\}$. (3) **Strong convexity criterion**: the Hessian matrix $g_{ij}(v) = \frac{1}{2}\frac{\partial^2 F^2}{\partial v^i v^j}(v)$ is positive definite for non-zero $v$.

A differentiable manifold $\mathcal{Z}$ equipped with a Finsler metric is called a **Finsler manifold**.

Finsler geometry can be seen as an extension of Riemannian geometry, since the requirements for defining a metric are less restrictive.

**Proposition 2.2.** Let $G$ be a stochastic Riemannian metric. Then, the function $F_z : \mathcal{T}_z\mathcal{Z} \to \mathbb{R} : u \to \|u\|_F$ defines a Finsler metric, but it is not induced by a Riemannian metric.

**Proof.** If $F$ was induced by a Riemannian metric, then this metric would be defined as: $f_z : \mathbb{R}^q \times \mathbb{R}^q \to \mathbb{R}_+ : (v_1, v_2) \to \mathbb{E}\left[\sqrt{v_1^\top G v_2}\right]^2$. Since a Riemannian metric is an inner product, it should be symmetric, positive, definite and bilinear. Here, we can see that $f_x$ is not bilinear, so $f_z$ is not a Riemannian metric. However, we can prove that $F_z : \mathbb{R}^q \to \mathbb{R} : v \to \mathbb{E}\left[\sqrt{v^\top G v}\right]$ is positive, homogeneous, smooth and strongly convex, and so $F_z$ is a Finsler metric (Shen & Shen, 2016, Definition 2.1). For the full proof, see Section B.1. $\qquad\square$

So far, we have assumed that $f$ is an immersion and a stochastic process. If we consider $f$ to be a **Gaussian Process** in particular, the Finsler norm can be rewritten in a closed form expression.

**Proposition 2.3.** Let $f$ be a Gaussian process and $J$ its Jacobian, with $J \sim \mathcal{N}(\mathbb{E}[J], \Sigma)$. The Finsler norm can be written as:

$$F_z : \mathcal{T}_z\mathcal{Z} \to \mathbb{R}_+ : \|v\|_F := v \to \sqrt{2}\sqrt{v^\top \Sigma v}\frac{\Gamma(\frac{D}{2} + \frac{1}{2})}{\Gamma(\frac{D}{2})}{}_1F_1\left(-\frac{1}{2}, \frac{D}{2}, -\frac{\omega}{2}\right),$$

with ${}_1F_1$ as the confluent hypergeometric function of the first kind and $\omega = (v^\top \Sigma v)^{-1}(v^\top \mathbb{E}[J]^\top \mathbb{E}[J]v)$.

**Proof.** We suppose that $f$ is a Gaussian process, and so is its Jacobian. $G$ follows a non-central Wishart distribution: $G = J^\top J \sim \mathcal{W}_q(D, \Sigma, \Sigma^{-1}\mathbb{E}[J]^\top \mathbb{E}[J])$. $v^\top G v$ is a scalar and also follows a non-central Wishart distribution: $v^\top G v \sim \mathcal{W}_1(D, \sigma, \omega)$, with $\sigma = v^\top \Sigma v$ and $\omega = (v^\top \Sigma v)^{-1}(v^\top \mathbb{E}[J]^\top \mathbb{E}[J]v)$ (Kent & Muirhead, 1984, Definition 10.3.1). The square-root of a non-central Wishart distribution follows a non-central Nakagami distribution (Hauberg, 2018b). Then, by construction, the stochastic norm $\|\cdot\|_G$ follows a non-central Nakagami distribution. The expectation of this distribution is known, and it has a closed-form expression. $\qquad\square$

The confluent hypergeometric function of the first kind, also known as the Kummer function, is a special function that is defined as the solution of a specific second-order linear differential equation. The term $\omega$ appears from the non-central Wishart distribution. When $\omega$ is non-zero, the distribution of the Jacobian shifts away from the origin, and $\omega$ represents the magnitude and the direction of this shift, balanced by the correlation between the variables. In Section 3.4, to prove our results in high-dimensions, we will assume that our manifold $\mathcal{Z}$ is bounded, and so is $\omega$.

## 3 Comparison of Riemannian and Finsler metrics

### 3.1 Theoretical comparison

In geometry, we need to define a metric (a norm) to compute functionals. In Riemannian geometry, the metric is conveniently obtained by constructing an inner product. Because of its bilinearity, the inner product greatly simplifies subsequent computations, but it is also restrictive. A generalisation of Riemannian geometry can be obtained by relaxing this assumption. Instead of defining a metric as an inner product, we can define

the metric as a norm[2]. Relaxing this assumption was studied by Finsler (1918), who gave his name to this discipline.

Finsler geometry is similar to Riemannian geometry without the bilinear assumption. Most of the functionals (curve length and curve energy) are defined similarly to those obtained in Riemannian geometry. However, the volume measure is different, and there are at least two definitions of volume measure used in Finsler geometry: the Busemann-Hausdorff volume and the Holmes-Thomson volume measure (Wu, 2011). In this paper, we decided to focus on the Busemann-Hausdorff definition (Definition 3.1), which is more intuitive and easier to derive. If the Finsler metric is a Riemannian metric, the definition of volume naturally coincides with the Riemannian volume measure.

In Figure 3, a Busemann-Hausdorff and the Riemannian volume measures have been computed for the same set of data points. A Gaussian process has been trained to fit data representing a pinwheel projected onto a sphere.

**Definition 3.1.** For a given point $z$ on the manifold, we define the **Finsler indicatrix** as the set of vectors in the tangent space such that the Finsler metric is equal to: $\{v \in \mathcal{T}_z \mathcal{Z} | F_z(v) = 1\}$). We call $\mathbb{B}^n(1)$ the Euclidean unit ball, and $\text{vol}(\cdot)$ the standard Euclidean volume. In local coordinates $(e^1, \cdots, e^d)$ on a Finsler manifold $\mathcal{M}$, the **Busemann-Hausdorff volume** form is defined as $dV_F = V_F(z)e^1 \wedge \cdots \wedge e^d$, with:

$$V_F(z) = \frac{\text{vol}(\mathbb{B}^n(1))}{\text{vol}(\{v \in \mathcal{T}_z \mathcal{Z} | F_z(v) < 1\})}.$$

In the definition above, we introduce the notion of *indicatrix*. An indicatrix is a way to represent the distortion induced by the metric on a unit circle. If our metric is euclidean, we will only have a linear transformation between the latent and the observational spaces, and the indicatrix would still be a circle. Because the Riemannian metric is quadratic, it will always generate an ellipse in the latent space. The Finsler indicatrix, however, would have a convex, even asymmetrical, shape. This difference can be observed in the indicatrix-field represented in Figure 2: The Finsler indicatrices in purple can have almost rectangular shape, while the Riemannian indicatrices, in orange, are ellipses.

There are also a few observations to note in Figure 2. First, in the area of low predictive variance (where data points lie in the latent space), the Finsler and Riemannian indicatrices are alike. This follows from the preceding comment that the metrics diverge by a variance term. If our mapping $f$ was deterministic, both metrics would agree. Second, for every point, the Riemannian indicatrices are always contained by the Finslerian ones, illustrating Proposition 3.1 on our absolute bounds in the following section.

## 3.2 Absolute bounds on the Finsler metric

The Finsler norm is upper bounded with the Riemannian norm obtained from the expected metric tensor. It is also lower bounded:

**Proposition 3.1.** We define $\alpha = 2\left(\frac{\Gamma(\frac{D}{2}+\frac{1}{2})}{\Gamma(\frac{D}{2})}\right)^2$. The Finsler norm: $\|\cdot\|_F$ is bounded by two norms, $\|\cdot\|_{\alpha\Sigma}$ and $\|\cdot\|_R$, induced by the two respective Riemannian metric tensors: the covariance tensor $\alpha\Sigma_z$ and the expected metric tensor $\mathbb{E}[G_z]$.

$$\forall (z,v) \in \mathcal{Z} \times \mathcal{T}_z Z: \ \|v\|_{\alpha\Sigma} \leq \|v\|_F \leq \|v\|_R$$

---

[2]A norm only needs to be definite and satisfies the triangular inequality, but is not necessary symmetric. This means that, for a vector $v$, we can have a non reversible Finsler metric: $F_x(v) \neq F_x(-v)$. Intuitively, this means that the path used to connect two points would be different depending on the starting point. This asymmetric property becomes valuable when studying the geometry of anisotropic media (Markvorsen, 2016), for example. In our case, our Finsler metric is reversible.

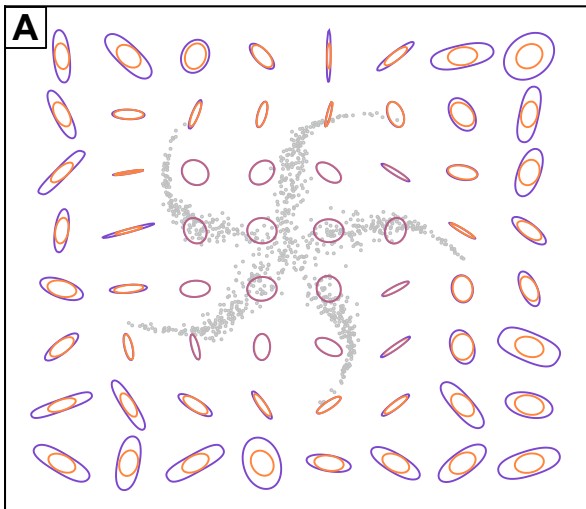 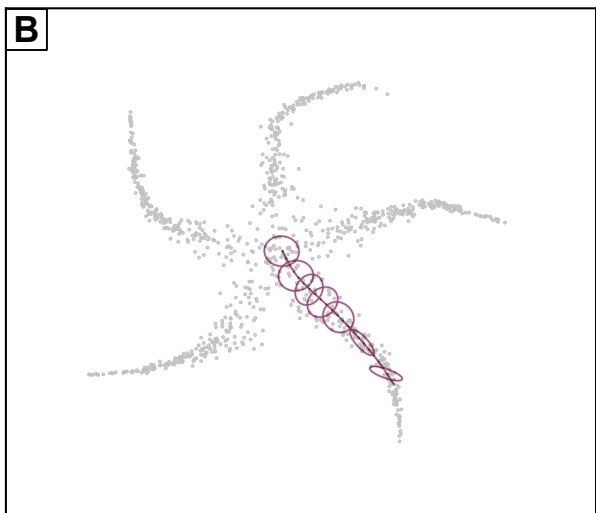

Figure 2: Indicatrice field over the latent space of the pinwheel data (in grey) representing the Riemannian (in orange) and Finslerian (in purple) metrics (See Section C). (A) The indicatrices are computed over a grid in the latent space. (B) The indicatrices are computed along a geodesic: the Riemannian and Finslerian metrics coincide.

**Proof.** The proof can be sketched as follows: the upper bound $\|v\|_F \leq \|v\|_R$, also rewritten as: $\mathbb{E}[\sqrt{v^\top G v}] \leq \sqrt{v^\top \mathbb{E}[G] v}$, is obtained by applying Jensen's inequality, knowing that the square root $x \to \sqrt{x}$ is a concave function. The lower bound $\|v\|_{\alpha\Sigma} \leq \|v\|_F$, rewritten as $\sqrt{v^\top \alpha\Sigma v} \leq \mathbb{E}[\sqrt{v^\top G v}]$, is obtained using the closed form expression of the Finsler function. $\qquad \square$

The result is illustrated in Figure 4 (lower right). Four metric tensors $(G_1, G_2, G_3, G_4)$, each following a non-central Wishart distribution with a specific mean and covariance matrix, have been computed. For each of them, we have drawn the indicatrices $(\{v \in \mathcal{T}_z\mathcal{Z} \mid \|v\| = 1\})$ induced by the norms: $\|\cdot\|_F$, $\|\cdot\|_R$ and $\|\cdot\|_{\alpha\Sigma}$. As expected, we can notice that the $\alpha\Sigma$-indicatrix contains the Finsler indicatrix, itself containing $R$-indicatrix.

By bounding the Finsler metric, we are able to bound their respective functionals:

> **Corollary 3.1.** The length, the energy and the Busemann-Hausdorff volume of the Finsler metric are bounded respectively by the Riemannian length, energy and volume of the covariance tensor $\alpha\Sigma$ (noted $L_{\alpha\Sigma}, E_{\alpha\Sigma}, V_{\alpha\Sigma}$) and the expected metric $\mathbb{E}[G]$ (noted $L_R, E_R, V_R$):
>
> $$\forall z \in \mathcal{Z}, \ L_{\alpha\Sigma}(z) \leq L_F(z) \leq L_R(z)$$
> $$E_{\alpha\Sigma}(z) \leq E_F(z) \leq E_R(z)$$
> $$V_{\alpha\Sigma}(z) \leq V_F(z) \leq V_R(z)$$

**Proof.** From Proposition 3.1, we need to integrate each term of the inequality to obtain the length and the energy. The volume is less trivial, since we use the Busemann-Hausdorff definition for measuring $V_F$. We have to place ourselves in hyperspherical coordinates, and show that the Finsler indicatrix is still bounded. $\qquad \square$

### 3.3 Relative bounds on the Finsler metric

**Proposition 3.2.** Let $f$ be a stochastic immersion. $f$ induces the stochastic norm $\|\cdot\|_G$, defined in Section 2. The relative difference between the Finsler norm $\|\cdot\|_F$ and the Riemmanian norm $\|\cdot\|_R$ is:

$$0 \leq \frac{\|v\|_R - \|v\|_F}{\|v\|_R} \leq \frac{\mathrm{Var}\left[\|v\|_G^2\right]}{2\mathbb{E}\left[\|v\|_G^2\right]^2}.$$

**Proof.** This proposition is a direct application of the Sharpened Jensen's inequality (Liao & Berg, 2019). $\square$

The previous proposition is valid for any stochastic immersion. We can see that the metrics become equal when the ratio of the variance over the expectation shrinks to zero. This happens in two cases: when the variance converges to zero, which is similar to having a deterministic immersion, and when the number of dimensions increases. The latter case is investigated below for a Gaussian process [3].

**Proposition 3.3.** Let $f$ be a Gaussian process. We note $\omega = (v^\top \Sigma v)^{-1}(v^\top \mathbb{E}[J]^\top \mathbb{E}[J]v)$, with $J$ the jacobian of $f$, and $\Sigma$ the covariance matrix of $J$.
The relative ratio between the Finsler norm $\|\cdot\|_F$ and the Riemmanian norm $\|\cdot\|_R$ is:

$$0 \leq \frac{\|v\|_R - \|v\|_F}{\|v\|_R} \leq \frac{1}{D + \omega} + \frac{\omega}{(D + \omega)^2}.$$

**Proof.** $v^\top G v$ follows a one-dimension non-central Wishart distribution: $v^\top G_z v \sim \mathcal{W}_1(D, \sigma, \omega)$, with $\sigma = v^\top \Sigma v$ and $\omega = (v^\top \Sigma v)^{-1}(v^\top \mathbb{E}[J]^\top \mathbb{E}[J]v)$. We use the theorem of the moments to obtain both the expectation and the variance, which leads us to the result. $\square$

**Corollary 3.2.** When $f$ is a Gaussian Process, the relative ratio between the length, the energy and the volume of the Finsler norm (noted $L_F, E_F, V_F$) and the Riemannian norm (noted $L_R, E_R, V_R$) is:

$$0 \leq \frac{L_R(z) - L_F(z)}{L_R(z)} \leq \max_{v \in \mathcal{T}_z \mathcal{Z}} \left\{ \frac{1}{D + \omega} + \frac{\omega}{(D + \omega)^2} \right\}$$

$$0 \leq \frac{E_R(z) - E_F(z)}{E_R(z)} \leq \max_{v \in \mathcal{T}_z \mathcal{Z}} \left\{ \frac{2}{D + \omega} + \frac{1 + 2\omega}{(D + \omega)^2} + \frac{2\omega}{(D + \omega)^3} + \frac{\omega^2}{(D + \omega)^4} \right\}$$

$$0 \leq \frac{V_R(z) - V_F(z)}{V_R(z)} \leq 1 - \left( 1 - \max_{v \in \mathcal{T}_z \mathcal{Z}} \left\{ \frac{1}{D + \omega} + \frac{\omega}{(D + \omega)^2} \right\} \right)^q$$

**Proof.** We directly use Proposition 3.3. To obtain the inequalities with the lengths and the energies, we first multiply all the terms by the Riemannian metric, and we integrate every term. To obtain the inequality with the volume, similarly to Corollary 3.1, we place ourselves in hyperspherical coordinates and bound the radius of the Finsler indicatrix. $\square$

In Figure 3, we can compare the volume measures obtained from the Riemannian and Finsler metrics, and in particular, their ratio in the top right image. When the metrics are computed next to the data points in area where the variance is very low, we can see that the ratio of the volume measure is at the order of magnitude $10^{-4}$. Further away from the data points, the variance increases and so does the difference between the Riemannian and Finsler volume measures.

---

[3] Interestingly, the term $\mathbb{E}[\nu]^2/\mathrm{Var}[\nu]$, when $\nu$ follows a central Nakagami distribution, is called a shape parameter. It has been introduced by Nakagami himself to study the intensity of fading in radio wave propagation (Nakagami, 1960). When $\nu := \sqrt{\xi}$ with $\xi \sim \mathcal{W}_1(\Sigma, D)$, then $m = D/2$. This is a particular result obtained from Proposition 3.3, when $\mathbb{E}[J] = 0$.

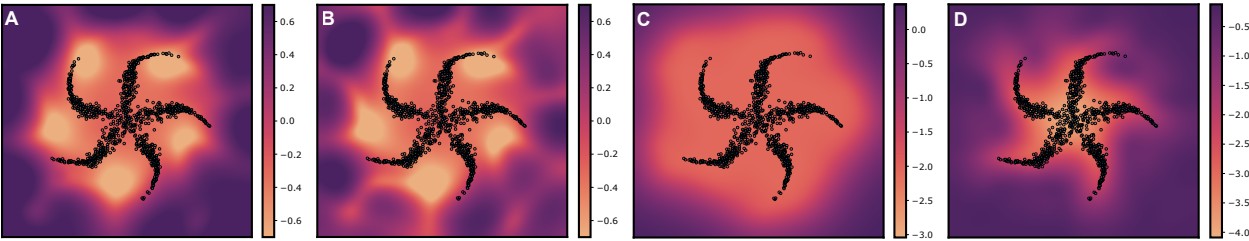

Figure 3: Difference of volume for data embedded in the latent space. (A) Riemannian volume measure, (B) Finslerian (Busemann-Hausdorff) volume measure, (C) Variance of the Gaussian process, (D) Ratio between the Riemannian and Finslerian volume: $(V_R(z) - V_F(z))/V_R(z)$. All heatmaps are computed in logarithm scale.

### 3.4 Results in high dimensions

Proposition 3.3 and Corollary 3.2 indicate that the metrics become similar when the dimension ($D$) of the observational space increases. If we assume that the latent space is a bounded manifold, the metrics converge to each other at a rate of $\mathcal{O}\left(\frac{1}{D}\right)$, as do their functionals.

We assume that the latent manifold is bounded. Then, we can deduce that (1) the term $\omega$, which represents the non-centrality of the data, does not grow faster than the number of dimensions (See lemma B.2.2, in Section B.2.3) and (2) we that the metrics are finite.

**Corollary 3.3.** Let $f$ be a Gaussian Process. In high dimensions, we have:

$$\frac{L_R(z) - L_F(z)}{L_R(z)} = \mathcal{O}\left(\frac{1}{D}\right)$$

$$\frac{E_R(z) - E_F(z)}{E_R(z)} = \mathcal{O}\left(\frac{1}{D}\right)$$

$$\frac{V_R(z) - V_F(z)}{V_R(z)} = \mathcal{O}\left(\frac{q}{D}\right)$$

And, when $D$ converges toward infinity: $L_R \underset{+\infty}{\sim} L_F$, $E_R \underset{+\infty}{\sim} E_F$ and $V_R \underset{+\infty}{\sim} V_F$.

**Proof.** This result follows from Corollary 3.2, assuming the latent manifold is bounded. □

**Corollary 3.4.** Let $f$ be a Gaussian Process. In high dimensions, the relative ratio between the Finsler norm $\|\cdot\|_F$ and the Riemmanian norm $\|\cdot\|_R$ is:

$$\frac{\|v\|_R - \|v\|_F}{\|v\|_R} = \mathcal{O}\left(\frac{1}{D}\right)$$

And, when $D$ converges toward infinity: $\forall v \in \mathcal{T}_z\mathcal{Z}$, $\|v\|_R \underset{+\infty}{\sim} \|v\|_F$.

**Proof.** Similarly, from Proposition 3.3, in a bounded manifold, both metrics converge to each other in high dimensions. □

## 4 Experiments

We want to illustrate cases where these metrics could be useful in practice for real-world data. For this, we use three datasets: a synthetic dataset (composed of data representing a pinwheel projected onto a sphere),

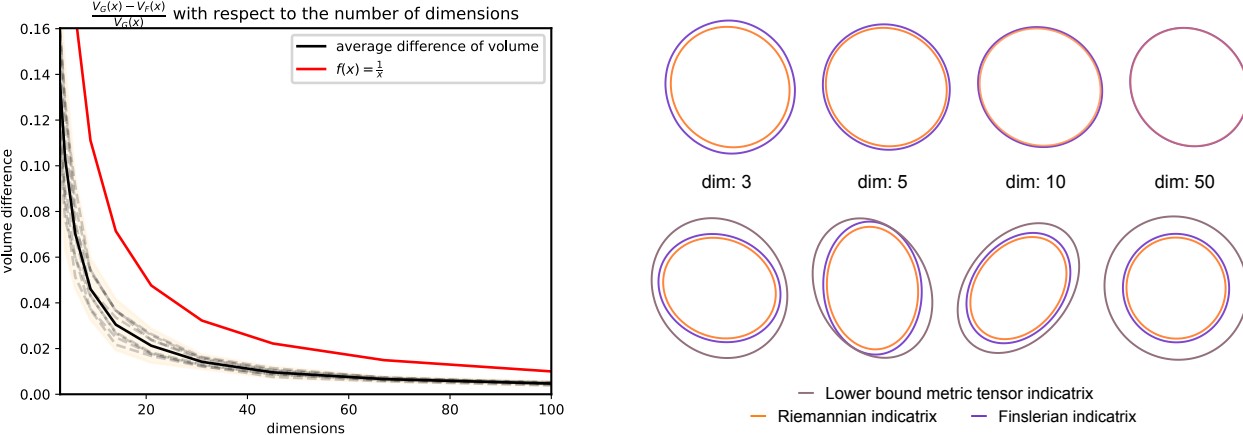

Figure 4: Left: Ratio of volumes $(V_R - V_F)/V_R$ decreasing with respect to the number of dimensions. The results were obtained from using a collection of matrices $\{G_i\}$ following a non-central Wishart distribution. Upper right: The Finsler and Riemannian indicatrices converge towards each other when increasing the number of dimensions. Lower right: Illustration of the absolute bounds in Proposition 3.1 with the $\alpha\Sigma$-indicatrices, Riemannian indicatrices and Finsler indicatrices.

a font dataset Campbell & Kautz (2014), and a dataset representing single-cells stages Guo et al. (2010). We trained a GP-LVM model to learn a 2d-manifold. From the optimised Gaussian process, we can access the Riemannian and Finsler metric, and minimise their respective curve energies to obtain geodesics.

As we can see, the Finsler and Riemannian geodesics coincide in all cases. For all latent spaces (A.1. and B.1. in Figure 5, and in Figure 6), the heatmap represents the Riemannian volume measure in logarithm scale. The volume measure is low in the area of high density and high in the area of low density of data points.

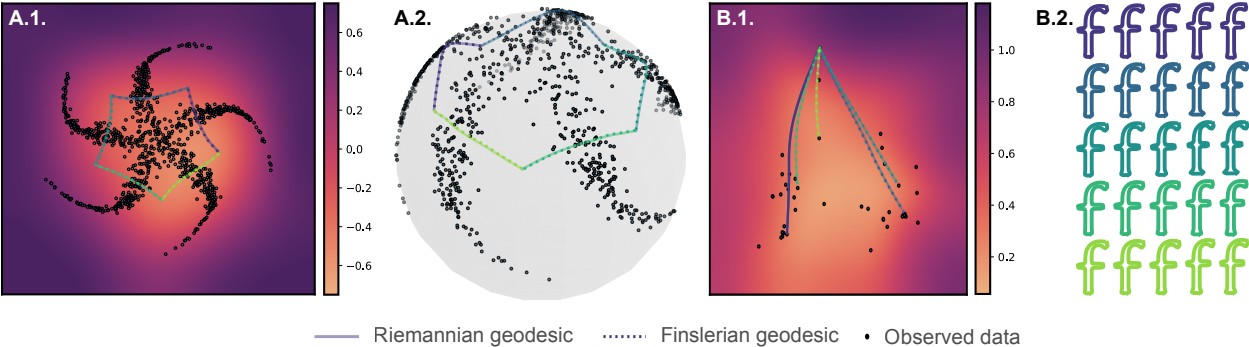

Figure 5: Geodesics computed for latent (A.1., B.1.) and observational (A.2., B.2.) spaces. (A) The dataset used was a pinwheel projected onto a sphere, as seen in A.2. (B) The dataset consists of the position of the markers parametrising the contour of the letter **f**.

## 5    Discussion

Generative models are often used to reduce data dimension in order to better understand the mechanisms behind the data generating process. We consider the general setting where the mapping from latent variables to observations is driven by a smooth stochastic process, and the sample mappings span Riemannian manifolds. The Riemannian geometry machinery has already been used in the past to explore the latent space.

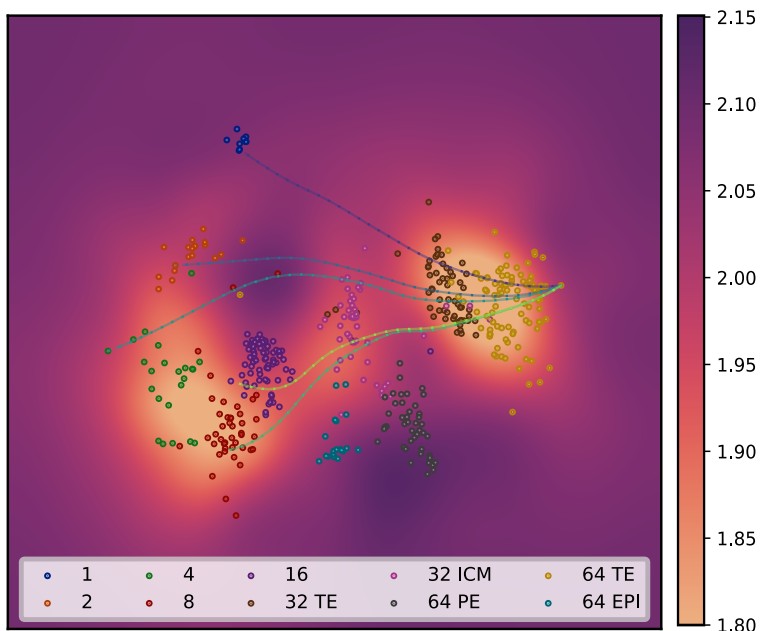

Figure 6: qPCR data

In this paper, we have shown how curves and volumes can be identified by defining the length of a latent curve as its expected length measured in the observation space. This is a natural extension of classical differential geometric constructions to the stochastic realm. Surprisingly, we have shown that this does not give rise to a Riemannian metric over the latent space, even if sample mappings do. Rather, the latent representation naturally becomes equipped with a Finsler metric, implying that stochastic manifolds, such as those spanned by Latent Variable Models (LVMs), are inherently more complex than their deterministic counterparts.

The Finslerian view of the latent representation gives us a suitable general solution to explore a random manifold, but it does not immediately translate into a practical computational tool. As Riemannian manifolds are better understood computationally than Finsler manifolds, we have raised the question: How good an approximation of the Finsler metric can be achieved by a Riemannian metric? The answer turns out to be: quite good. We have shown that as data dimension increases, the Finsler metric becomes increasingly Riemannian. Since LVMs are most commonly applied to high-dimensional data (as this is where dimensionality reduction carries value), we have justification for approximating the Finsler metric with a Riemannian metric such that computational tools become more easily available. In practice we find that geodesics under the Finsler the Riemannian metric are near identical, except in regions of high uncertainty.

**Notations**

| | |
|---|---|
| $\mathcal{Z}, \mathcal{X}$ | Smooth differentiable latent ($\mathcal{Z}$) and data ($\mathcal{X}$) manifold, |
| $f$ | A stochastic immersion $f : \mathcal{Z} \subset \mathbb{R}^q \to \mathcal{X} \subset \mathbb{R}^D$, |
| $J$ | Jacobian of the stochastic function $f$, |
| $G$ | Stochastic metric tensor defined as the pullback metric through $f$: $G = J^\top J$, |
| $\mathcal{T}_z \mathcal{Z}$ | Tangent space of the manifold $\mathcal{Z}$ at a point $z$, |
| $\Sigma$ | IF $f$ is a Gaussian process, then $J \sim \prod_{i=1}^{D} \mathcal{N}(\mu_i, \Sigma)$, |
| $\|\cdot\|_G$ | Stochastic induced norm: $\|v\|_G \coloneqq \sqrt{v^\top G v}$, |
| $\|\cdot\|_R$ | Riemannian induced norm: $\|v\|_R \coloneqq \sqrt{v^\top \mathbb{E}[G] v} \coloneqq \sqrt{g(v,v)}$, |
| $\|\cdot\|_F$ | Finsler norm: $\|v\|_F \coloneqq \mathbb{E}[\sqrt{v^\top G v}] = F(v)$, |
| $L_R, E_R, V_R$ | Length, energy and volume obatined from the Riemannian induced norm $\|\cdot\|_R$, |
| $L_F, E_F, V_F$ | Length, energy and Busemann Hausdorff volume obtained from the Finsler norm $\|\cdot\|_F$. |

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

# A  A primer on Geometry

The main purpose of the paper is to define and compare two legitimate metrics to compute the average length between random points. Before going further, it's important to formally define the two metrics (Riemannian and Finsler metrics, respectively) which we do in sections A.2 and A.3. They are both constructed on topological manifolds, the definition of which is recalled in section A.1. We finally introduce the notion of random manifold in section A.4, which is the last notion needed to frame our problem of interest: which metric should we use to compute the average distance on a random manifold?

## A.1  Topological and differentiable manifolds

This section aims to define core concepts in differential geometry that will be used later to define Riemannian and Finsler manifolds. Recall that two topological spaces are called homeomorphic if there is a continuous bijection between them with continuous inverse.

> **Definition A.1.** A d-dimensional **topological manifold** $\mathcal{M}$ is a second-countable Hausdorff topological space such that every point has an open neighbourhood homeomorphic to an open subset of $\mathbb{R}^d$.

Let $\mathcal{M}$ be a topological manifold. This means that for any $x \in \mathcal{M}$ there is an open neighbourhood $U_x$ of $x$ and a homeomorphism $\phi_{U_x} : U_x \to \mathbb{R}^d$ onto an open subset of $\mathbb{R}^d$. Suppose that $x, y \in \mathcal{M}$ are such that $U_x \cap U_y \neq \emptyset$, let $U = U_x$, $V = U_y$ and consider the so-called coordinate change map

$$\phi_V \circ \phi_{U|\phi_U(U \cap V)}^{-1} : \phi_U(U \cap V) \to \mathbb{R}^d.$$

We call $\mathcal{M}$ together with an open cover $\{U_x\}_{x \in \mathcal{M}}$ as above a **differentiable** or **smooth** manifold if the coordinate maps are infinitely differentiable.

Beyond these technical definitions, one can imagine a differentiable manifold as a well-behaved smooth surface that possesses *locally* all the topological properties of a Euclidean space. All the manifolds in this paper are assumed to be differentiable and connected manifolds.

> **Definition A.2.** We also define, for a differentiable manifold $\mathcal{M}$, the **tangent space** $\mathcal{T}_x M$ as the set of all the tangent vectors at $x \in \mathcal{M}$, and the **tangent bundle** $\mathcal{T}\mathcal{M}$ the disjoint union of all the tangent spaces: $\mathcal{T}\mathcal{M} = \underset{x \in \mathcal{M}}{\cup} \mathcal{T}_x M$.

So far, we have only defined topological and differential properties of manifolds. In order to compute geometric quantities, we need to equip those with a metric that helps us derive useful quantities such as lengths, energies and volumes. A metric is a scalar valued function that is defined for each point on the topological manifold and takes as inputs one or two vectors (depending on the type of metric) from the tangent space at the specific point. Such a function can either be defined as a scalar product between two vectors, this is the case of a Riemannian metric or, in the case of a Finsler metric, it is defined similarly to the norm of a vector. We will formally define these metrics and highlight their differences in the following sections.

## A.2  Riemannian manifolds

> **Definition A.3.** Let $\mathcal{M}$ be a manifold. A **Riemannian metric** is a map assigning at each point $x \in \mathcal{M}$ a scalar product $G(\cdot, \cdot) : \mathcal{T}_x M \times \mathcal{T}_x M \to \mathbb{R}$, with $G$ a positive definite bilinear map, which is smooth with respect to $x$. A smooth manifold equipped with a Riemannian metric is called a **Riemannian manifold**. We usually express the metric as a symmetric positive definite matrix $G$, where we have for two vectors $u, v \in \mathcal{T}_x M$: $G(u, v) = \langle u, v \rangle_G = u^\top G v$. We further define the induced **norm**: $v \in \mathcal{T}_x M, \|v\|_G = \sqrt{G(v, v)}$.

The Riemannian metric here can either refer to the scalar product $G$ itself, or the associated metric tensor $G$.

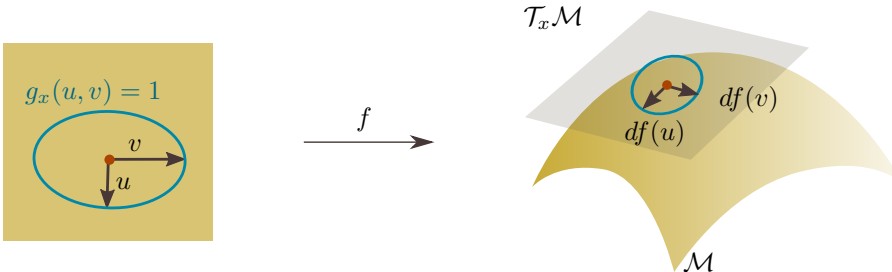

Figure 7: $f$ is an immersion that maps a low dimensional manifold to a high dimensional manifold $\mathcal{M}$. On $\mathcal{M}$, a tangent plane $\mathcal{T}_x\mathcal{M}$ is draw at $x$. The indicatrice of the Euclidean metric is plotted in blue. When this metric is pulled-back through $f$, the low dimensional space is now equipped with the pullback metric $g$, which is a Riemannian metric by definition. The vectors $df(u)$ and $df(v)$ are called the push-forwards of the vectors $u$ and $v$ through $f$.

**Definition A.4.** We consider a curve $\gamma(t)$ and its derivative $\dot\gamma(t)$ on a Riemannian manifold $\mathcal{M}$ equipped with the metric $g$. Then, we define the **length of the curve**:

$$L_G(\gamma) = \int \left\| \dot\gamma(t) \right\|_G dt = \int \sqrt{g_t(\dot\gamma(t), \dot\gamma(t))}\, dt,$$

where $g_t = g_{\gamma(t)}$. Locally length-minimising curves between two connecting points are called **Geodesics**.

**Definition A.5.** The **curve energy** is defined as:

$$E_G(\gamma) = \int \left\| \dot\gamma(t) \right\|_G^2 dt = \int g_t(\dot\gamma(t), \dot\gamma(t))\, dt.$$

There are two interesting properties to note about the length of a curve and the curve energy. First, the length is parametrisation invariant: for any bijective smooth function $\eta$ on the domain of $\gamma$ we have that $L_G(\gamma \circ \eta) = L_G(\gamma)$. We also say the Riemannian metric gives us intrinsic coordinates to compute the length. Secondly, for a given curve $\gamma$, we have: $L_G(\gamma)^2 \leq 2E_G(\gamma)$. Because of the invariance of the curve, when we aim to minimise it, a solver can find an infinite number of solutions. On the other hand, the curve energy is convex and will lead to a unique solution. Thus, to obtain a geodesic, instead of solving the corresponding ODE equations, or directly minimising lengths, it is easier in practice to minimise the curve energy, as a minimal energy gives a minimal length.

The Riemannian metric also provides us with an infinitesimal volume element that relates our metric $G$ to an orthonormal basis, the same way the Jacobian determinant accommodate for a change of coordinates in the change of variables theorem.

**Definition A.6.** In local coordinates $(e^1, \cdots, e^d)$, the **volume form** of the Riemannian manifold $\mathcal{M}$, equipped with the metric tensor $G$, is defined as: $dV_G = V_G(x)e^1 \wedge \cdots \wedge e^d$, with:

$$V_G(x) = \sqrt{\det(G)}.$$

**Remark.** *The symbol $\wedge$ represents the wedge product and it is used to manipulate differential k-forms. Here, the basis vectors $(e^1, \cdots, e^d)$ form a d-dimensional parallelepiped $(e^1 \wedge \cdots \wedge e^d)$ with unit volume.*

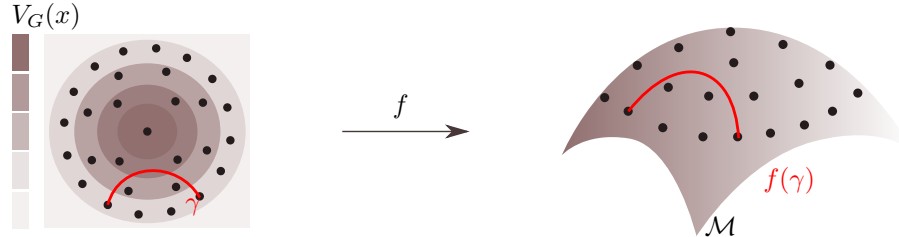

Figure 8: Once the low dimensional manifold is equipped with a metric that captures the inherent structure of the high dimensional manifold, we can compute a geodesic $\gamma$, by minimising the energy functional between two points. The geodesic $f(\gamma)$ will be the shortest path between two points on the manifold $\mathcal{M}$. The volume measure $V_G$ can also be used to integrate functions over regions of the manifold, as we would do in the Euclidean space. It can also be linked to the density of the data: if the data points are uniformly distributed over the high-dimensional manifold, in the low-dimensional manifold, a low volume would correspond to a high density of data. It is a useful way to give more information about the distribution of the data.

### A.3 Finsler manifolds

Finsler geometry is often described as an extension of Riemannian geometry, since the metric is defined in a more general way, lifting the quadratic constraint. In particular, the norm of a Riemmanian metric is a Finsler metric, but the converse is not true.

**Definition A.7.** Let $F : \mathcal{T}\mathcal{M} \to \mathbb{R}_+$ be a continuous non-negative function defined on the tangent bundle $\mathcal{T}\mathcal{M}$ of a differentiable manifold $M$. We say that $F$ is a **Finsler metric** if, for each point $x$ of $\mathcal{M}$ and $v$ on $\mathcal{T}_x M$, we have:

1. Positive homogeneity: $\forall \lambda \in \mathbb{R}_+$, $F(\lambda v) = \lambda F(v)$.

2. Smoothness: $F$ is a $C^\infty$ function on the slit tangent bundle $\mathcal{T}\mathcal{M} \setminus \{0\}$.

3. Strong convexity criterion: the Hessian matrix $g_{ij}(v) = \frac{1}{2}\frac{\partial^2 F^2}{\partial v^i v^j}(v)$ is positive definite for non-zero $v$.

A differentiable manifold $\mathcal{M}$ equipped with a Finsler metric is called a **Finsler manifold**.

Here, it is worth noting that, for a given point in the manifold, the Finsler metric is defined with only one vector in the tangent space, while the Riemannian metric is defined with two vectors. Moreover, from the previous definition, we can deduce that the metric is:

1. Positive definite: for all $x \in \mathcal{M}$ and $v \in \mathcal{T}_x M$, $F(v) \geq 0$ and $F(v) = 0$ if and only if $v = 0$.

2. Subadditive: $F(v + w) \leq F(v) + F(w)$ for all $x \in \mathcal{M}$ and $v, w \in \mathcal{T}_x M$.

We say that $F$ is a Minkowski norm on each tangent space $\mathcal{T}_x M$. Furthermore, if $F$ satisfies the reversibility property: $F(v) = F(-v)$, it defines a norm on $\mathcal{T}_x M$ in the usual sense.

Similarly to Riemannian geometry, lengths, energies and volumes can be defined directly from the Finsler metric:

**Definition A.8.** We consider a curve $\gamma$ and its derivative $\dot{\gamma}$ on a Finsler manifold $\mathcal{M}$ equipped with the metric $F$. We define the **length of the curve** as follows:

$$L_F(\gamma) = \int F(\dot{\gamma}(t))dt.$$

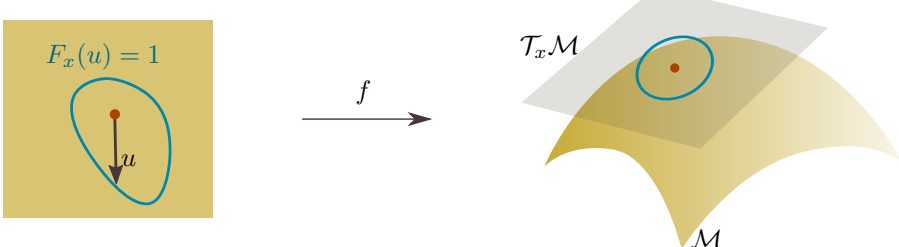

Figure 9: $f$ is an immersion that maps a low dimensional manifold to a high dimensional manifold $\mathcal{M}$. On $\mathcal{M}$, a tangent plane $\mathcal{T}_x\mathcal{M}$ is draw at $x$. Compared to the Riemannian manifold, the Finsler indicatrix, which represents the all the vectors $u \in \mathcal{T}_x\mathcal{M}$ such that $F_x(u) = 1$, is not necessarily an ellipse. It can be asymmetric if the metric is asymmetric itself. It is always convex.

**Definition A.9.** The **curve energy** is defined as: $E_F(\gamma) = \int F(\dot{\gamma}(t))^2 dt$.

Not only are the definitions strikingly similar, they also share the same properties. The curve length is also invariant under reparametrisation, and upper bounded by the curve energy. Computing geodesics on a manifold is reduced to a variational optimisation problem. These propositions are proved in detail in Lemmas B.1.4 and B.1.5, in the appendix.

In Riemannian geometry, the volume measure defined by the metric is unique. In Finsler geometry, different definitions of the volume exist, and they all coincide with the Riemannian volume element when the metric is Riemannian. The most common choices of volume forms are the Busemann-Hausdorff measure and the Holmes-Thompson measure. According Wu (2011), depending on the Finsler metric and the topological manifold, some choices seem more legitimate than others. In this paper, we decided to only focus on the Busemann-Hausdorff volume, as its definition is the most commonly used and leads to easier derivations. We will later show that in high dimensions, our Finsler metric converges to a Riemannian metric, and thus, the results obtained for the Busemann-Hausdorff volume measure are also valid for the Holmes-Thomson volume measure.

**Definition A.10.** For a given point $x$ on the manifold, we define the **Finsler indicatrix** as the set of vectors in the tangent space such that the Finsler metric is equal to one: $\{v \in \mathcal{T}_xM | F(v) = 1\}$. We denote the Euclidean unit ball in $\mathbb{R}^d$ by $\mathbb{B}^d(1)$ and for measurable subsets $S \subseteq \mathbb{R}^d$ we use vol($S$) to denote the standard Eulcidean volume of $S$. In local coordinates $(e^1, \cdots, e^d)$ on a Finsler manifold $\mathcal{M}$, the **Busemann-Hausdorff volume** form is defined as $dV_F = V_F(x)e^1 \wedge \cdots \wedge e^d$, with:

$$V_F(x) = \frac{\text{vol}(\mathbb{B}^d(1))}{\text{vol}(\{v \in \mathcal{T}_xM | F(v) < 1\})}.$$

We can interpret the volume as the ratio between the euclidean ball, and a convex ball whose radius is defined as a unit Finsler metric. If the Finsler metric is replaced by a Riemannian metric, the volume of the indicatrix will be an ellipsoid whose semi-axis are equal to the inverse of the squareroot of the metric's eigenvalues. The Finsler volume then reduces to the definition of the Riemannian volume.

### A.4   Random manifolds

So far, we have only considered deterministic data points lying on a manifold. If we consider our data to be random variables, we will need to define the associated random metric and manifold.

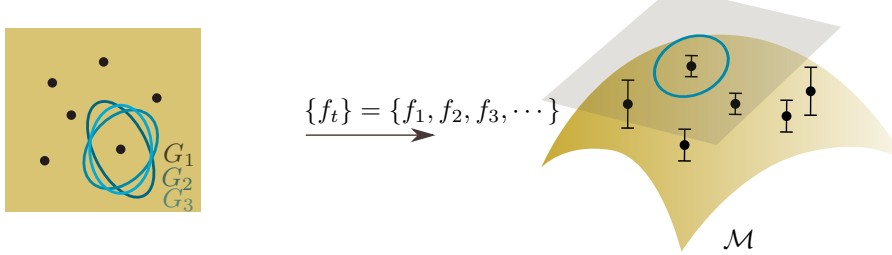

$$\{f_t\} = \{f_1, f_2, f_3, \cdots\}$$

Figure 10: Usually the immersion $f$ would be deterministic. In the case of most generative models, where $f$ is described by a GP-LVM, or the decoder of a VAE, the immersion is stochastic. The pullback metric is stochastic de facto.

As said previously, if we have a function $f : \mathbb{R}^q \to \mathbb{R}^D$ that parametrises a manifold, then we can construct a Riemannian metric $G = J_f^\top J_f$, with $J_f$ the Jacobian of the function $f$. In the previous cases, we assumed $f$ to be a deterministic function, and so is the metric. We construct a stochastic Riemannian metric in the same way, with $f$ being a stochastic process. A stochastic process $f : \mathbb{R}^q \to \mathbb{R}^D$ is a random map in the sense that samples of the process are maps from $\mathbb{R}^q$ to $\mathbb{R}^D$ (the so-called sample paths of the process).

**Definition A.11.** A stochastic process $f : \mathbb{R}^q \to \mathbb{R}^D$ is smooth if the sample paths of $f$ are smooth. We call a smooth process f a **stochastic immersion** if the Jacobian matrix of its sample paths has full rank everywhere. We can then define the **stochastic Riemannian metric** $G = J_f^\top J_f$.

The terms *stochastic* and *random* are used interchangeably. The definition of the stochastic immersion is fairly important, as it means that its Jacobian is full rank. Since the Jacobian is full rank, the random metric $G$ is positive definite, a necessary condition to define a Riemannian metric. Another definition of a stochastic Riemannian metric would be the following:

**Definition A.12.** A **stochastic Riemannian metric** on $\mathbb{R}^q$ is a matrix-valued random field on $\mathbb{R}^q$ whose sample paths are Riemannian metrics. A **stochastic manifold** is a differentiable manifold equipped with a stochastic Riemannian metric.

Any matrice drawn from this stochastic metric would be a proper Riemannian metric. When using the random Riemannian metric on two vectors $u, v \in \mathcal{T}_x M$, $G(u, v) = u^\top G v$ is a random variable, but both $u, v$ are deterministic vectors. From this definition, it follows that the length, the energy and the also volume are random variables.

| Object | Riemann | Finsler |
|---|---|---|
| metric | $g : \mathcal{T}_x M \times \mathcal{T}_x M \to \mathbb{R}$ | $F : \mathcal{T} M \to \mathbb{R}_+$ |
| length structure | $L_G(\gamma) = \int \sqrt{g_t(\dot{\gamma}(t), \dot{\gamma}(t))} \, dt$ | $L_F(\gamma) = \int F(\dot{\gamma}(t)) \, dt$ |
| energy structure | $E_G(\gamma) = \int g_t(\dot{\gamma}(t), \dot{\gamma}(t)) \, dt$ | $E_F(\gamma) = \int F(\dot{\gamma}(t))^2 \, dt$ |
| volume element | $V_G(x) = \sqrt{|\det\{G\}|}$ | $V_F(x) = \mathrm{vol}(\mathbb{B}^n(1))/\mathrm{vol}(\{v \in \mathcal{T}_x M | F(x, v) < 1\})$ Busemann-Hausdorff volume measure |

Table 1: Comparison of Riemannian and Finsler metrics.

# B Proofs

One of the main challenges of this paper is to find coherent notations while respecting the tradition of two geometric fields. In Riemannian geometry, we principally use a metric, noted $g_p : \mathcal{T}_p\mathcal{M} \times \mathcal{T}_p\mathcal{M} \to \mathbb{R}_+$, that is defined as an inner product and thus can induce a norm, but is not a norm. In Finsler geometry, we call intercheangeably Finsler function, Finsler metric or Finsler norm, the norm traditionally noted $F : \mathcal{T}\mathcal{M} \to \mathbb{R}_+$, with $F_p(u) := F(p, u)$ defined at a point $p \in \mathcal{M}$ for a vector $u \in \mathcal{T}_p\mathcal{M}$. We will assume that all our metric are always defined for a specific point $z$ (or $p$) on our manifold $\mathcal{Z}$ (or $\mathcal{M}$), and so we will just drop this index. The following notations will be used:

| | |
|---|---|
| Stochastic pullback metric tensor | $G = J_f^\top J_f$ |
| Stochastic pullback metric | $\tilde{g} : (u, v) \to u^\top G u$ |
| Expected Riemannian metric | $g : (u, v) \to u^\top \mathbb{E}[G] v$ |
| Stochastic pullback induced norm | $\|\cdot\|_G : u \to \sqrt{u^\top G u}$ |
| Expected Riemannian induced norm | $\|\cdot\|_R : u \to \sqrt{u^\top \mathbb{E}[G] u} := \sqrt{g(u, u)}$ |
| Finsler metric | $\|\cdot\|_F : u \to \mathbb{E}[\sqrt{u^\top G u}] := F(u)$ |

## B.1 Finslerian geometry of the expected length

In this section, we will always let $f : \mathbb{R}^q \to \mathbb{R}^D$ be a stochastic immersion, $J_f$ its Jacobian, and $G = J_f^\top J_f$ a metric tensor. We will first prove that the function $F : \mathcal{T}M \to \mathbb{R} : v \to \mathbb{E}\left[\sqrt{v^\top G v}\right]$ is a Finsler metric. Then, for the specific case where $J_f$ follows a non-central normal distribution, the Finsler metric $F$ defined as the expected length follows a non-central Nakagami distribution and can be expressed in closed form.

To prove that the function $F$ is indeed a Finsler metric, we will need to verify the criteria above, among them the strong convexity criterion is less trivial to prove than the others. It will be detailed in Lemma B.1.3. Strong convexity means that the Hessian matrix $\frac{1}{2}\text{Hess}(F(v)^2) = \frac{1}{2}\frac{\partial^2 F^2}{\partial v^i v^j}(v)$ is strictly positive definite for non-negative $v$. This matrix, when $F$ is a Finsler function, is also called the fundamental form and plays an important role in Finsler geometry. To prove the strong convexity criterion, we will need the full expression of the fundamental form, detailed in Lemma B.1.1.

**Lemma B.1.1.** *The Hessian matrix $\frac{1}{2}\text{Hess}(F(v)^2)$ of the function $F(v) = \mathbb{E}\left[\sqrt{v^\top G v}\right]$ is given by*

$$\frac{1}{2}\text{Hess}(F(v)^2) = \mathbb{E}\left[(v^\top G v)^{\frac{1}{2}}\right] \mathbb{E}\left[(v^\top G v)^{-\frac{1}{2}}G - (v^\top G v)^{-\frac{3}{2}}Gvv^\top G\right] + \mathbb{E}\left[(v^\top G v)^{-\frac{1}{2}}G\right]^2 vv^\top.$$

**Proof.** Let $G$ be a random positive definite symmetric matrix and define $g : \mathbb{R}^q \to \mathbb{R} : v \mapsto \sqrt{v^\top G v}$, where $v$ is considered a column vector. We would like to know the different derivatives of $g$ with respect to $v$. We name by default $J_g$ and $H_g$, its Jacobian and Hessian matrix. Using the chain rule, we have: $J_g = (v^\top G v)^{-\frac{1}{2}}v^\top G$ and $H_g = (v^\top G v)^{-\frac{1}{2}}G - (v^\top G v)^{-\frac{3}{2}}(Gvv^\top G)$.

For the rest of the proof, we need to show that derivatives and expectation values commute.

Using the Fubini theorem, we can show that tha derivatives and the expectation values commute.

For $F : \mathbb{R}^q \to \mathbb{R} : v \mapsto \mathbb{E}[\sqrt{v^\top G v}]$,

$$\text{Hess}(F) = \mathbb{E}[H_g] = \mathbb{E}\left[(v^\top G v)^{-\frac{1}{2}}G - (v^\top G v)^{-\frac{3}{2}}Gvv^\top G\right]$$

$$\nabla F = \mathbb{E}[J_g] = \mathbb{E}[(v^\top G v)^{-\frac{1}{2}}Gv].$$

We now consider the function $h : \mathbb{R}^q \to \mathbb{R} : v \mapsto \mathbb{E}[\sqrt{v^\top G v}]^2 = F(v)^2$. Using the chain rule and changing the order of expectation and derivatives, we have its Hessian

$$H_h = 2F \cdot \text{Hess}[F] + 2\nabla F^\top \nabla F = 2\mathbb{E}[g]\mathbb{E}[H_g] + 2\mathbb{E}[J_g]^\top \mathbb{E}[J_g].$$

Finally, replacing $J_g$ and $H_g$ previously obtained in this expression, we conclude:

$$\frac{1}{2}H_h(x,v) = \mathbb{E}\left[(v^\top Gv)^{\frac{1}{2}}\right]\mathbb{E}\left[(v^\top Gv)^{-\frac{1}{2}}G - (v^\top Gv)^{-\frac{3}{2}}Gvv^\top G\right] + \mathbb{E}\left[(v^\top Gv)^{-\frac{1}{2}}G\right]^2 vv^\top.$$

$\square$

**Remark.** *Before going further, it's important to note that $G = J_f^\top J_f$ is a random matrix that is positive definite: it is symmetric by definition and has full rank. The later statement is justified by the assumption that the stochastic process $f : \mathbb{R}^q \to \mathbb{R}^D$ is an immersion, then $J_f$ is full rank.*

**Lemma B.1.2.** *The function $F(v) = \mathbb{E}\left[\sqrt{v^\top Gv}\right]$ is:*

1. *positive homogeneous: $\forall \lambda \in \mathbb{R}_+$, $F(\lambda v) = \lambda F(v)$*

2. *smooth: $F(v)$ is a $C^\infty$ function on the slit tangent bundle $\mathcal{TM} \setminus \{0\}$*

**Proof.** 1) Let $\lambda \in \mathbb{R}$, then we have: $F(\lambda v) = \mathbb{E}\left[\sqrt{\lambda^2 v^\top Gv}\right] = |\lambda|\left[\sqrt{v^\top Gv}\right]$.

2) The multivariate function: $\mathbb{R}^q \setminus \{0\} \to \mathbb{R}_+^* : v \to v^\top Gv$ is $C^\infty$ and strictly positive, since $G = J_f^\top J_f$ is positive definite. The function $\mathbb{R}_+^* \to \mathbb{R}_+^* : x \to \sqrt{x}$ is also $C^\infty$. Finally, $\mathbb{R}_+^* \to \mathbb{R}_+^* : x \to \mathbb{E}[x]$ is by definition differentiable. By composition, $F(v)$ is a $C^\infty$ function on the slit tangent bundle $\mathcal{TM} \setminus \{0\}$. $\square$

**Lemma B.1.3.** *The function $F(v) = \mathbb{E}\left[\sqrt{v^\top Gv}\right]$ satisfies the strong convexity criterion.*

**Proof.** Proving that F satisfies the strong convexity criterion is equivalent to show that the Hessian matrix $H = \frac{1}{2}\text{Hess}(F(v)^2)$ is strictly positive definite. Thus, we need to prove that $\forall w \in \mathbb{R}^q \setminus \{0\}, w^\top Hw > 0$. According to Lemma B.1.1, because the expectation is a positive function, it's straightforward to see that $\forall w \in \mathbb{R}^q \setminus \{0\}, w^\top Hw \geq 0$. The tricky part of this proof is to show that $w^\top Hw > 0$. This can be obtained if one of the terms ($F \cdot \text{Hess}(F)$ or $\nabla F^\top \nabla F$) is strictly positive.

First, let's decompose $H$ as the sum of matrices: $H = F\text{Hess}(F) + \nabla F^\top \nabla F$ (Lemma B.1.1), with:

$$F \cdot \text{Hess}(F) = \mathbb{E}\left[(v^\top Gv)^{\frac{1}{2}}\right]\mathbb{E}\left[(v^\top Gv)^{-\frac{3}{2}}\left((v^\top Gv)G - Gv(Gv)^\top\right)\right],$$

$$\nabla F^\top \nabla F = \mathbb{E}\left[(v^\top Gv)^{-\frac{1}{2}}G\right]^2 vv^\top.$$

We will study two cases: when $w \in \text{span}(v)$, and when $w \notin \text{span}(v)$. We will always assume that $v \neq 0$, and so by definition: $F(v) > 0$.

Let $w \in \text{span}(v)$. We will show that $w^\top \nabla F^\top \nabla Fw > 0$. We have $w = \alpha v, \alpha \in \mathbb{R}$. Because F is 1-homogeneous and using Euler theorem, we have: $\nabla F(v)v = F(v)$. Then $(\alpha v)^\top \nabla F^\top \nabla F(\alpha v) = \alpha^2 F^2$, and $\alpha^2 F(v)^2 > 0$.

Let $w \notin \text{span}(v)$. F being a scalar function, we have: $w^\top F\text{Hess}[F]w = Fw^\top \text{Hess}[F]w$. We would like to show that: $w^\top \text{Hess}[F]w > 0$. The strategy is the following: if we prove that the kernel of $\text{Hess}[F]$ is equal to the $\text{span}(v)$, then $w \notin \text{span}(v)$ is equivalent to say that $w \notin \ker(\text{Hess}[F])$ and we can conclude that: $w^\top \text{Hess}[F]w > 0$. Let's prove $\text{span}(v) \in \ker(\text{Hess}(F))$. We know that $\text{Hess}(F)v = 0$, since F is 1-homogeneous, so we have $\text{span}(v) \in \ker(\text{Hess}(F))$. To obtain the equality, we just need to prove that the dimension of the kernel is equal to 1. Let $z \in \text{span}(v^\top G)^\top$, which is $(Gv)^Tz = 0$. We have $\dim(\text{span}(v^\top M)) = 1$, and thus: $\dim(\text{span}(v^\top G)^\top) = q - 1$. Furthermore, $z^\top \text{Hess}[F]z = z^\top \mathbb{E}\left[M(v^\top Mv)^{-\frac{1}{2}}\right]z > 0$, so we can deduce that $\dim(\text{im}(\text{Hess}[F])) = q - 1$. Using the Rank-Nullity theorem, we conclude that $\dim(\ker(\text{Hess}(F))) = q - \dim(\text{im}(\text{Hess}[F])) = 1$, which concludes the proof.

In conclusion, $\forall w \in \mathbb{R}^q \setminus \{0\}, w^\top \frac{1}{2}\text{Hess}(F(v)^2)w > 0$. The function $F$ satisfies the strong convexity criterion. $\square$

**Proposition 2.2.** Let $G$ be a stochastic Riemannian metric. Then, the function $F_z : \mathcal{T}_z\mathcal{Z} \to \mathbb{R} : u \to \|u\|_F$ defines a Finsler metric, but it is not induced by a Riemannian metric.

**Proof.** Let's define F as a Riemannian metric: $F : \mathbb{R}^q \times \mathbb{R}^q \to \mathbb{R} : (v_1, v_2) \to \mathbb{E}\left[\sqrt{v_1^\top G v_2}\right]$. If $F$ were a Riemannian metric, then it would be bilinear, which is clearly not the case. Thus, $F$ is not a Riemannian metric. According to Lemma B.1.2 and Lemma B.1.2, $F$ is a Finsler metric. $\square$

**Proposition 2.3.** Let $f$ be a Gaussian process and $J$ its Jacobian, with $J \sim \mathcal{N}(\mathbb{E}[J], \Sigma)$. The Finsler norm can be written as:

$$F_z : \mathcal{T}_z\mathcal{Z} \to \mathbb{R}_+ : \|v\|_F := v \to \sqrt{2}\sqrt{v^\top \Sigma v}\frac{\Gamma(\frac{D}{2} + \frac{1}{2})}{\Gamma(\frac{D}{2})} {}_1F_1\left(-\frac{1}{2}, \frac{D}{2}, -\frac{\omega}{2}\right),$$

with ${}_1F_1$ as the confluent hypergeometric function of the first kind and $\omega = (v^\top \Sigma v)^{-1}(v^\top \mathbb{E}[J]^\top \mathbb{E}[J]v)$.

**Proof.** The objective of the proof is to show that, if the Jacobian $J_f$ follows a non-central normal distribution, then, $\forall v \in \mathbb{R}^q$, the expectation $\mathbb{E}[v^\top J_f^\top J_f v]$ will follow a non-central Nakagami distribution. This is a particular case of the derivation of moments of non-central Wishart distributions, previously shown and studied by Kent & Muirhead (1984); Hauberg (2018b).

By hypothesis, $J_f$ follows a non-central normal distribution: $J_f \sim \mathcal{N}(\mathbb{E}[J], I_D \otimes \Sigma)$. Then, $G = J_f^\top J_f$ follows a non-central Wishart distribution: $G \sim \mathcal{W}_d(D, \Sigma, \Sigma^{-1}\mathbb{E}[J]^\top \mathbb{E}[J])$. According to (Kent & Muirhead, 1984, Theorem 10.3.5.), $v^\top G v$ will also follow a non-central Wishart distribution: $v^\top G v \sim \mathcal{W}_1(D, v^\top \Sigma v, \omega)$, with: $\omega = (v^\top \Sigma v)^{-1}(v^\top \mathbb{E}[J]^\top \mathbb{E}[J]v)$.

To compute $\mathbb{E}[\sqrt{v^\top G v}]$, we shall look at the derivation of moments. (Kent & Muirhead, 1984, Theorem 10.3.7.) states that: if $X \sim \mathcal{W}_q(D, \Sigma, \Omega')$, with $q \leq D$, then $\mathbb{E}[(\det(X))^k] = (\det\{\Sigma\})^k 2^{qk}\frac{\Gamma_q(\frac{D}{2}+k)}{\Gamma_q(\frac{D}{2})} {}_1F_1(-k, \frac{D}{2}, -\frac{1}{2}\Omega')$. We directly apply the theorem to our case, knowing that $v^\top G v$ is a scalar term, so $\det(v^\top G v) = v^\top G v$, $q = 1$, and $k = \frac{1}{2}$:

$$\|v\|_F := \mathbb{E}[\sqrt{v^\top G v}] = \sqrt{2}\sqrt{v^\top \Sigma v}\frac{\Gamma(\frac{D}{2} + \frac{1}{2})}{\Gamma(\frac{D}{2})} {}_1F_1(-\frac{1}{2}, \frac{D}{2}, -\frac{1}{2}\omega)$$

$\square$

**Lemma B.1.4.** *The length of a curve using a Finsler metric is invariant by reparametrisation.*

**Proof.** The proof is similar to the one obtained on a Riemannian manifold (Lee (2013), Proposition 13.25), where we make use of the homogeneity property of the Finsler metric.

Let $(\mathcal{M}, F)$ be a Finsler manifold and $\gamma : [a, b] \to \mathcal{M}$ a piecewise smooth curve segment. We call $\tilde{\gamma}$ a reparametrisation of $\gamma$, such that $\tilde{\gamma} = \gamma \circ \phi$ with $\phi : [c, d] \to [a, b]$ a diffeomorphism. We want to show that $L_F(\gamma) = L_F(\tilde{\gamma})$.

$$L_F(\tilde{\gamma}) = \int_c^d F(\dot{\tilde{\gamma}}(t))\, dt = \int_c^d F(\frac{d}{dt}(\gamma \circ \phi(t)))\, dt$$
$$= \int_{\phi^{-1}(a)}^{\phi^{-1}(b)} |\dot{\phi}(t)|F(\dot{\gamma} \circ \phi(t))\, dt = \int_a^b F(\dot{\gamma}(t))\, dt = L_F(\gamma)$$

$\square$

**Lemma B.1.5.** *If a curve globally minimizes its energy on a Finsler manifold, then it also globally minimizes its length and the Finsler function F of the velocity vector along the curve is constant.*

**Proof.**  The curve energy and the curve length are defined as: $E_F(\gamma) = \int_0^1 F^2(\dot{\gamma}(t))dt$ and $L_F(\gamma) = \int_0^1 F(\dot{\gamma}(t))dt$, with $\gamma : [0,1] \to \mathbb{R}^d$. Let's define $f$ and $g$ two real-valued functions such that: $f : \mathbb{R} \to \mathbb{R} : t \mapsto F(\dot{\gamma}(t))$ and $g : \mathbb{R} \to \mathbb{R} : t \mapsto 1$. Applying Cauchy-Schwartz inequality, we directly obtain:

$$\left( \int_0^1 F(\dot{\gamma}(t))dt \right)^2 \leq \int_0^1 F(\dot{\gamma}(t))^2 dt \cdot \int_0^1 1^2 dt, \quad \text{which means:} \quad L_F(\gamma)^2 \leq E_F(\gamma).$$

The equality is obtained exactly when the functions $f$ and $g$ are proportional, hence, when the Finsler function is constant. $\qquad\square$

## B.2  Comparison of Riemannian and Finsler metrics

We have defined both a Riemannian ($g : (v_1, v_2) \to v_1^\top \mathbb{E}[G]v_2$) and a Finsler ($F : (x,v) \to \mathbb{E}[\sqrt{v^\top G v}]$) metric, in the hope to compute the average length between two points on a random manifold created by the random field $f$: $G = J_f^\top J_f$. The main idea of this section is to better compare those two metrics and in what extend they differ in terms of length, energy and volume. From now on, $f : \mathbb{R}^q \to \mathbb{R}^D$ will always be defined as a stochastic non-central gaussian process. Its Jacobian $J_f$ also follows a non-central gaussian distribution, $G = J_f^\top J_f$ a non-central Wishart distribution, and $F : (x,v) = \mathbb{E}[\sqrt{v^\top G v}]$ a non-central Nakagami distribution (Proposition 2.3). The Finsler metric can be written in closed form.

In section B.2.1, we will see that the Finsler metric is upper and lower bounded by two Riemannian tensors (Proposition 3.1), and we can deduce an upper and lower bound for the length, the energy and the volume (Corollary 3.1). Then, in section B.2.2, we will show that the relative difference between the Finsler norm and the Riemannian induced norm is always positive and upper bounded a term that is inversely proportional to the number of dimensions $D$ (Proposition 3.3). Similarly, we will deduce the same for the length, the energy and the volume (Corollary 3.2). From this last results, we can directly conclude in section B.2.3 that both metrics are equal in high dimensions (Corollary 3.4). A possible interpretation is that in high dimensions the data distribution obtained on those manifolds becomes more and more concentrated around the mean, reducing the variance term to zero. The manifold becoming deterministic, both metrics become equal.

**Remark.** *Most of the following proofs will be a bit technical, as they rely on the closed form expression of the non-central Nakagami distribution. Once proving the main propositions, obtaining the corollaries is straightforward. While we do not have closed form expression of the indicatrix, we will show that it's a monotoneous function which can upper and lower bounded.*

### B.2.1  Bounds on the Finsler metric

**Proposition 3.1.** We define $\alpha = 2\left( \frac{\Gamma(\frac{D}{2} + \frac{1}{2})}{\Gamma(\frac{D}{2})} \right)^2$. The Finsler norm: $\|\cdot\|_F$ is bounded by two norms, $\|\cdot\|_{\alpha\Sigma}$ and $\|\cdot\|_R$, induced by the two respective Riemannian metric tensors: the covariance tensor $\alpha\Sigma_z$ and the expected metric tensor $\mathbb{E}[G_z]$.

$$\forall (z, v) \in \mathcal{Z} \times \mathcal{T}_z Z : \|v\|_{\alpha\Sigma} \leq \|v\|_F \leq \|v\|_R$$

**Proof.**  Let's first recall that the Finsler function can be written as:

$$\|v\|_F := F(v) = \sqrt{2}\sqrt{v^\top \Sigma v} \frac{\Gamma(\frac{D}{2} + \frac{1}{2})}{\Gamma(\frac{D}{2})} \cdot {}_1F_1(-\frac{1}{2}, \frac{D}{2}, -\frac{1}{2}\omega).$$

The confluent hypergeometric function is defined as: ${}_1F_1(a, b, z) = \sum_{k=0}^\infty \frac{(a)_k}{(b)_k} \frac{z^k}{k!}$, with $(a)_k$ and $(b)_k$ being the Pochhammer symbols. Note that, despite their confusing notation, they are defined as rising factorials. By definition, we have: $\frac{(a)_k}{(b)_k} = \frac{\Gamma(a+k)}{\Gamma(b+k)} \frac{\Gamma(b)}{\Gamma(a)}$. We can use the Kummer transformation to obtain:

$_1F_1(a, b, -z) = e^{-z}{}_1F_1(b - a, b, z)$. Replacing $a = -\frac{1}{2}$, $b = \frac{D}{2}$ and $z = \frac{1}{2}\omega$, we finally get:

$$F(v) = \sqrt{2}\sqrt{v^\top \Sigma v} \cdot e^{-z} \sum_{k=0}^{\infty} \frac{\Gamma(\frac{D}{2} + \frac{1}{2} + k)}{\Gamma(\frac{D}{2} + k)} \frac{z^k}{k!}.$$

1) Let's show that: $\forall v \in \mathcal{T}_x M : \sqrt{v^\top \alpha \Sigma v} \leq F(v)$, with $\alpha = 2\left(\frac{\Gamma(\frac{D}{2} + \frac{1}{2})}{\Gamma(\frac{D}{2})}\right)^2$.

The Pochhammer symbole is defined as $(x)_k = x(x+1)\dots(x+k-1) = \frac{\Gamma(x+k)}{\Gamma(x)}$. For $x \in \mathbb{R}_+^*$, we have: $(x)_k \leq (x + \frac{1}{2})_k$. Thus, $\frac{\Gamma(\frac{D}{2} + k)}{\Gamma(\frac{D}{2})} \leq \frac{\Gamma(\frac{D}{2} + \frac{1}{2} + k)}{\Gamma(\frac{D}{2} + \frac{1}{2})}$. The Gamma function being strictly positive on $\mathbb{R}_+$, we obtain:

$$\frac{\Gamma(\frac{D}{2} + \frac{1}{2})}{\Gamma(\frac{D}{2})} \leq \frac{\Gamma(\frac{D}{2} + \frac{1}{2} + k)}{\Gamma(\frac{D}{2} + k)}$$

$$\sqrt{2}\sqrt{v^\top \Sigma v} \frac{\Gamma(\frac{D}{2} + \frac{1}{2})}{\Gamma(\frac{D}{2})} \cdot e^{-z} \sum_{k=0}^{\infty} \frac{z^k}{k!} \leq \sqrt{2}\sqrt{v^\top \Sigma v} \cdot e^{-z} \sum_{k=0}^{\infty} \frac{\Gamma(\frac{D}{2} + \frac{1}{2} + k)}{\Gamma(\frac{D}{2} + k)} \frac{z^k}{k!}$$

$$\sqrt{2}\frac{\Gamma(\frac{D}{2} + \frac{1}{2})}{\Gamma(\frac{D}{2})}\sqrt{v^\top \Sigma v} \leq \sqrt{2}\sqrt{v^\top \Sigma v} \cdot e^{-z} \sum_{k=0}^{\infty} \frac{\Gamma(\frac{D}{2} + \frac{1}{2} + k)}{\Gamma(\frac{D}{2} + k)} \frac{z^k}{k!}$$

$$\sqrt{v^\top \alpha \Sigma v} \leq F(v).$$

2) Let's show that: $\forall v \in \mathcal{T}_x M : F(v) \leq \sqrt{v^\top \mathbb{E}[G]v}$.

Wendel (1948) proved: $\frac{\Gamma(x+y)}{\Gamma(x)} \leq x^y$, for $x > 0$ and $y \in [0, 1]$. With $x = \frac{D}{2} + k$, $y = \frac{1}{2}$, we obtained $\frac{\Gamma(\frac{D}{2} + \frac{1}{2} + k)}{\Gamma(\frac{D}{2} + k)} \leq \sqrt{\frac{D}{2} + k}$, which leads to: $F(v) \leq \sqrt{2v^\top \Sigma v} \cdot e^{-z} \sum_{k=0}^{\infty} \sqrt{\frac{D}{2} + k}\frac{z^k}{k!}$.

Furthermore, $\sum_{k=0}^{\infty} e^{-z}\frac{z^k}{k!} = 1$ and the function $x \to \sqrt{\frac{D}{2} + x}$ is concave. Then by Jensen's inequality: $e^{-z} \sum_{k=0}^{\infty} \sqrt{\frac{D}{2} + k}\frac{z^k}{k!} \leq \sqrt{\frac{D}{2} + e^{-z} \sum_{k=0}^{\infty} \frac{z^k}{k!}k}$. Knowing that $\sum_{k=0}^{\infty} \frac{z^k}{k!} = ze^z$, we have: $e^{-z} \sum_{k=0}^{\infty} \sqrt{\frac{D}{2} + k}\frac{z^k}{k!} \leq \sqrt{\frac{D}{2} + z}$.
And with $z = \frac{\Omega}{2}$, we obtain: $F(v) \leq \sqrt{v^\top \Sigma(D + \Omega)v}$.

From (Kent & Muirhead, 1984, p. 442), the expectation of a non-central Wishart distribution ($G \sim \mathcal{W}_q(D, \Sigma, \Omega)$) is: $\mathbb{E}[G] = D\Sigma + \Sigma\Omega$. This finally leads to:

$$F(v) \leq \sqrt{v^\top \mathbb{E}[G]v}.$$

$\square$

**Remark.** *As a side note, the second part of the inequality $F(v) \leq \sqrt{v^\top \mathbb{E}[G]v}$ can be obtained using directly Proposition 3.2.*

**Corollary 3.1.** The length, the energy and the Busemann-Hausdorff volume of the Finsler metric are bounded respectively by the Riemannian length, energy and volume of the covariance tensor $\alpha\Sigma$ (noted $L_{\alpha\Sigma}, E_{\alpha\Sigma}, V_{\alpha\Sigma}$) and the expected metric $\mathbb{E}[G]$ (noted $L_R, E_R, V_R$):

$$\forall z \in \mathcal{Z}, \; L_{\alpha\Sigma}(z) \leq L_F(z) \leq L_R(z)$$
$$E_{\alpha\Sigma}(z) \leq E_F(z) \leq E_R(z)$$
$$V_{\alpha\Sigma}(z) \leq V_F(z) \leq V_R(z)$$

**Proof.** From Proposition 3.1, we have $\forall (x,v) \in \mathcal{M} \times \mathcal{T}_x M : \sqrt{h(v)} \leq F(v) \leq \sqrt{g(v)}$, with $h : v \to v^\top \alpha \Sigma v$ and $g : v \to v^\top \mathbb{E}[G] v$ Riemannian metrics. We also define the parametric curve: $\forall t \in \mathbb{R}, \gamma(t) = x$ and $\dot{\gamma}(t) = v$.

1) Let's show that $L_\Sigma(x) \leq L_F(x) \leq L_R(x)$. Because of the monotonicity of the Lebesgue integrals, we directly have: $\int \sqrt{h(\dot{\gamma}(t))} dt \leq \int F(\dot{\gamma}(t)) dt \leq \int \sqrt{g(\dot{\gamma}(t))} dt$.

2) Let's show that $E_\Sigma(x) \leq E_F(x) \leq E_R(x)$. Since all the functions are positive, we can raise them to the power two, and again, with the monotonicity of the Lebesgue integrals, we have: $\int h(\dot{\gamma}(t)) dt \leq \int F^2(\gamma(t), \dot{\gamma}(t)) dt \leq \int \sqrt{g(\dot{\gamma}(t))} dt$.

3) Let's show that $V_\Sigma(x) \leq V_F(x) \leq V_R(x)$. We write the vectors $v \in \mathcal{T}_x M$ in hyperspherical coordinates: $v = re$, with $r = \|v\|$ the radial distance and $e$ the angular coordinates. With $v = re$, we have: $r \cdot \sqrt{h(e)} \leq r \cdot F(e) \leq r \cdot g(e) \iff \sqrt{h(e)}^{-1} \geq F(e)^{-1} \geq \sqrt{g(e)}^{-1}$.

We want to identify an inequality between the indicatrices, noted $\mathrm{vol}(I_h)$, $\mathrm{vol}(I_g)$, $\mathrm{vol}(I_F)$, formed by the functions $h$, $g$ and $F$. Let's define: $r_g \sqrt{h(e)} = r_h \sqrt{g(e)} = r_F F(e) = 1$. For every angular coordinate $e$, we obtain: $r_h \geq r_F \geq r_g$. Intuitively, this means that the finsler indicatrix will always be bounded by the indicatrices formed by $h$ and $g$. The Busemann-Hausdorf volume of a function $f$ is defined as: $\sigma_B(f) = \mathrm{vol}(\mathbb{B}^n(1))/\mathrm{vol}(I_f)$, with $\mathrm{vol}(\mathbb{B}^n(1))$ the volume of the unit ball and $\mathrm{vol}(I_f)$ the volume of the indicatrix formed by $f$. The previous inequality and the definition of the Busemann-Hausdorff volume implies that: $\mathrm{vol}(I_h) \geq \mathrm{vol}(I_F) \geq \mathrm{vol}(I_g) \Rightarrow \sigma_B(h) \leq \sigma_B(F) \leq \sigma_B(g)$. The functions $g$ and $h$ being Riemannian, we have: $\sigma_B(h) = \sqrt{det(\alpha \Sigma)}$ and $\sigma_B(g) = \sqrt{det(\mathbb{E}[G])}$, which concludes the proof.

$\square$

### B.2.2 Relative bounds between the Finsler and the Riemannian metric

**Proposition 3.2.** Let $f$ be a stochastic immersion. $f$ induces the stochastic norm $\|\cdot\|_G$, defined in Section 2. The relative difference between the Finsler norm $\|\cdot\|_F$ and the Riemmanian norm $\|\cdot\|_R$ is:

$$0 \leq \frac{\|v\|_R - \|v\|_F}{\|v\|_R} \leq \frac{\mathrm{Var}\left[\|v\|_G^2\right]}{2\mathbb{E}\left[\|v\|_G^2\right]^2}.$$

**Proof.** We will directly use a sharpen version of Jensen's inequality obtained by Liao & Berg (2019): Let $X$ be a one-dimensional random variable with mean $\mu$ and $P(X \in (a,b)) = 1$, where $-\infty \leq a \leq b \leq +\infty$. Let $\phi$ a twice derivable function on $(a,b)$. We further define: $h(x,\mu) = \frac{\phi(x) - \phi(\mu)}{(x-\mu)^2} - \frac{\phi'(\mu)}{x-\mu}$. Then:

$$\inf_{x \in (a,b)} \{h(x,\mu)\} \mathrm{Var}[X] \leq \mathbb{E}[\phi(x)] - \phi(\mathbb{E}[x]) \leq \sup_{x \in (a,b)} \{h(x,\mu)\} \mathrm{Var}[X].$$

In our case, we will chose $\phi : z \to \sqrt{z}$ with $z$ a one-dimensional random variable defined as $z = v^\top G v$. $a = 0$, $b = +\infty$ and $\mu = \mathbb{E}[z]$. $h(z,\mu) = (\sqrt{z} - \sqrt{\mu})(z-\mu)^{-2} - (2(z-\mu)\sqrt{\mu})^{-1}$. Because its first derivative $\phi'$ is convex, the function $x \to h(x,\mu)$ is monotonically increasing. Thus:

$$\inf_{z \in (0,+\infty)} \{h(x,\mu)\} = \lim_{z \to 0} = -\frac{\sqrt{\mu}}{2\mu^2} \quad \text{and} \quad \sup_{z \in (0,+\infty)} \{h(x,\mu)\} = \lim_{z \to +\infty} = 0.$$

It finally gives:

$$-\frac{\sqrt{\mu}}{2\mu^2} \mathrm{Var}[z] \leq \mathbb{E}[\sqrt{z}] - \sqrt{\mathbb{E}[z]} \leq 0.$$

Replacing $\|v\|_F := F(v) = \mathbb{E}[\sqrt{z}]$ and $\|v\|_R := \sqrt{g(v)} = \sqrt{\mathbb{E}[z]} = \sqrt{\mu}$ concludes the proof. $\square$

**Lemma B.2.1.** *Let $z \sim \mathcal{W}_1(D, \sigma, \Omega)$ following a one-dimensional non-central Wishart distribution. Then:*

$$\frac{Var[z]}{2\mathbb{E}[z]^2} = \frac{1}{D + \Omega} + \frac{\Omega}{(D + \Omega)^2}$$

**Proof.** (Kent & Muirhead, 1984, Theorem 10.3.7.) states that if $z \sim \mathcal{W}_1(D, \sigma, \omega)$ then $\mathbb{E}[z^k] = \sigma^k 2^k \frac{\Gamma(\frac{D}{2}+k)}{\Gamma(\frac{D}{2})} {}_1F_1(-k, \frac{D}{2}, -\frac{1}{2}\Omega)$. In particular, for $k = 1$ and $k = 2$, we have ${}_1F_1(-1, b, c) = 1 - \frac{c}{b}$ and ${}_1F_1(-2, b, c) = 1 - \frac{2c}{b} + \frac{c^2}{b(b+1)}$. We also have $\frac{\Gamma(\frac{D}{2}+1)}{\Gamma(\frac{D}{2})} = \frac{D}{2}$ and $\frac{\Gamma(\frac{D}{2}+2)}{\Gamma(\frac{D}{2})} = \frac{D}{2}\left(\frac{D}{2}+1\right)$, which leads to: $\mathbb{E}[z] = \sigma(D + \Omega)$ and $\mathbb{E}[z^2] = \sigma^2(2\omega + 2(D + \omega) + (D + \omega)^2)$. Finally, we conclude:

$$\frac{\text{Var}[z]}{\mathbb{E}[z]^2} = \frac{\mathbb{E}[z^2]}{\mathbb{E}[z]^2} - 1 = \frac{2\omega}{(D + \omega)^2} + \frac{2}{D + \omega}.$$

$\square$

**Proposition 3.3.** Let $f$ be a Gaussian process. We note $\omega = (v^\top \Sigma v)^{-1}(v^\top \mathbb{E}[J]^\top \mathbb{E}[J]v)$, with $J$ the jacobian of $f$, and $\Sigma$ the covariance matrix of $J$.
The relative ratio between the Finsler norm $\|\cdot\|_F$ and the Riemmanian norm $\|\cdot\|_R$ is:

$$0 \le \frac{\|v\|_R - \|v\|_F}{\|v\|_R} \le \frac{1}{D + \omega} + \frac{\omega}{(D + \omega)^2}.$$

**Proof.** The result is directly obtained using Proposition 3.2 and Lemma B.2.1. $\square$

**Corollary 3.2.** When $f$ is a Gaussian Process, the relative ratio between the length, the energy and the volume of the Finsler norm (noted $L_F, E_F, V_F$) and the Riemannian norm (noted $L_R, E_R, V_R$) is:

$$0 \le \frac{L_R(z) - L_F(z)}{L_R(z)} \le \max_{v \in \mathcal{T}_z \mathcal{Z}}\left\{\frac{1}{D + \omega} + \frac{\omega}{(D + \omega)^2}\right\}$$

$$0 \le \frac{E_R(z) - E_F(z)}{E_R(z)} \le \max_{v \in \mathcal{T}_z \mathcal{Z}}\left\{\frac{2}{D + \omega} + \frac{1 + 2\omega}{(D + \omega)^2} + \frac{2\omega}{(D + \omega)^3} + \frac{\omega^2}{(D + \omega)^4}\right\}$$

$$0 \le \frac{V_R(z) - V_F(z)}{V_R(z)} \le 1 - \left(1 - \max_{v \in \mathcal{T}_z \mathcal{Z}}\left\{\frac{1}{D + \omega} + \frac{\omega}{(D + \omega)^2}\right\}\right)^q$$

**Proof.** Let's call $M = \max_{v \in \mathcal{T}_x M}\{\frac{\omega}{(D+\omega)^2} + \frac{1}{D+\omega}\}$.

From Proposition 3.3, we have:

$$0 \le \|v\|_R - \|v\|_F \le M \|v\|_R, \quad \text{with} \quad M = \max_{v \in \mathcal{T}_x M}\left\{\frac{1}{D + \omega} + \frac{\omega}{(D + \omega)^2}\right\}$$

1) By the monocity of the Lesbesgue integral, we can directly integrate the previous norms along a curve $\gamma$, which immediately leads to: $0 \le L_R(x) - L_F(x) \le M L_R(x)$.

2) Since all the functions are positive: $0 \le \|v\|_F \le \|v\|_R \le M\|v\|_R + \|v\|_F$ leads to: $\|v\|_F^2 \le \|v\|_R^2 \le M^2\|v\|_R^2 + 2M\|v\|_F\|v\|_R + \|v\|_F^2$, and replacing $\|v\|_F \le \|v\|_R$ in the right hand term: $\|v\|_F^2 \le \|v\|_R^2 \le (M^2 + 2M)\|v\|_R^2 + \|v\|_F^2$, and finally: $0 \le \|v\|_R^2 - \|v\|_F^2 \le (M^2 + 2M)\|v\|_R^2$. Again, by continuity of the Lebesgue integral, we directly obtain: $0 \le E_R(x) - E_F(x) \le (M^2 + 2M)E_R(x)$.

3) In order to compare the volume between the Finsler and the Riemannian metric, we need to compare the volume of their indicatrices, noted: $\text{vol}(I_g)$ and $\text{vol}(I_F)$ respectively. We write the vectors $v \in \mathcal{T}_x M$ in hypershperical coordinates, with $v = re$, $r = \|v\|$ the radial distance and $e$ the angular coordinates. The volume of the indicatrices obtained in dimension $q$ (dimension of the latent space) can be written as: $d^q V = r^{q-1} dr d\Phi$, with $\Phi$ defining the different angles. We will note $r_F$ and $r_g$ the radial distances of the Finsler and Riemann metrics such that: $\|v\|_F = r_F \|e\|_F = 1$ and $\|v\|_R = r_g \|e\|_R = 1$ obtained for a specific angle $e$.

$$\text{vol}(I_F) - \text{vol}(I_g) = \int_\Phi \left( \int_0^{r_f} r^{q-1} dr - \int_0^{r_g} r^{q-1} dr \right) d\Phi = \int_\Phi \frac{r_f^q}{q} \left( 1 - \left( \frac{r_g}{r_f} \right)^q \right) d\Phi \leq \int_\Phi \frac{r_f^q}{q} d\Phi \cdot \left( 1 - \left( \frac{r_g}{r_f} \right)^q \right),$$

and by definition: $\text{vol}(I_F) = \int_\Phi (r_f^q/q) d\Phi$. Furthermore, for a specific angle $e$, we have: $r_g/r_F = \sqrt{g(e)}/F(e) \geq 1 - M$, from Proposition 3.3. We have:

$$0 \leq \frac{\text{vol}(I_F) - \text{vol}(I_g)}{\text{vol}(I_F)} \leq \cdot 1 - \left( \frac{r_g}{r_f} \right)^q \leq 1 - (1 - M)^q,$$

and by the definition of the Busemann Hausdorff volume: $\frac{V_F(x) - V_G(x)}{V_F(x)} = \frac{\text{vol}(I_F) - \text{vol}(I_g)}{\text{vol}(I_F)}$, we conclude the proof.

$\square$

### B.2.3 Implications in High Dimensions

In this section, we want to show that the difference between the Finsler norm and the Riemannian induced norm, as well as their respective functionals, tend to zero at a rate of $\mathcal{O}(\frac{1}{D})$. We need to be sure that $\omega$ doesn't grow faster than $D$, in other terms: $\omega = \mathcal{O}(D)$. This can be obtained if we assume that every element of the expectation of Jacobian is upper bounded ($\exists m \in \mathbb{R}_+^*, \forall i, j \ \mathbb{E}[J_{ij}] \leq m$). This happens in at least two cases: (1) $\mathbb{E}[f]$ is somehow Lipschitz continuous; or (2) if $f$ is a Gaussian Process and its covariance is upper bounded. The latter case happens when the process is defined over a bounded domain.

**Lemma B.2.2.** *Our Finsler metric $v \to \mathbb{E}[\sqrt{v^\top G v}]$ is defined with $v^\top G v \sim \mathcal{W}_1(D, v^\top \Sigma v, \omega)$, and $\omega = (v^\top \Sigma v)^{-1}(v^\top \mathbb{E}[J]^\top \mathbb{E}[J] v)$.*

*If the Finsler manifold is bounded, then: $\omega \leq DM$, with $M \in \mathbb{R}_+$.*

**Proof.** By definition, $\Sigma$ does not depend on $D$. We assume the manifold is bounded, which means that every element of the expected Jacobian is upper bounded: $\mathbb{E}[J]_{ij} \leq m$, with $m \in \mathbb{R}_+^*$. We call $\sigma = v^\top \Sigma v \in \mathbb{R}_+^*$.

We have:

$$\omega = \sigma^{-1} \sum_{k=1}^D \sum_{i=1}^q \sum_{j=1}^q v_i \mathbb{E}[J]_{ki} \mathbb{E}[J]_{kj} v_j \leq \sigma^{-1} \sum_{k=1}^D m^2 \|v\|^2 \leq DM,$$

with $M = \sigma^{-1} m^2 \|v\|^2 \in \mathbb{R}_+^*$, and M does not depend on D. $\square$

**Corollary 3.3.** Let $f$ be a Gaussian Process. In high dimensions, we have:

$$\frac{L_R(z) - L_F(z)}{L_R(z)} = \mathcal{O}\left( \frac{1}{D} \right)$$

$$\frac{E_R(z) - E_F(z)}{E_R(z)} = \mathcal{O}\left( \frac{1}{D} \right)$$

$$\frac{V_R(z) - V_F(z)}{V_R(z)} = \mathcal{O}\left( \frac{q}{D} \right)$$

And, when $D$ converges toward infinity: $L_R \underset{+\infty}{\sim} L_F$, $E_R \underset{+\infty}{\sim} E_F$ and $V_R \underset{+\infty}{\sim} V_F$.

**Proof.** From Corollary 3.2, we directly obtained the results in high dimensions.

We assume that our latent space is bounded, then by B.2.2, we have: $0 \leq \omega \leq MD$, with $M \in \mathbb{R}+$.

For the length, we have:
$$\frac{L_G(x) - L_F(x)}{L_G(x)} \leq \max_{v \in \mathcal{T}_x M} \left\{ \frac{1}{D+\omega} + \frac{\omega}{(D+\omega)^2} \right\}$$
$$\leq \frac{1+M}{D}$$

For the energy functional, we have:

$$\frac{E_G(x) - E_F(x)}{E_G(x)} \leq \max_{v \in \mathcal{T}_x M} \left\{ \frac{2}{D+\omega} + \frac{1+2\omega}{(D+\omega)^2} + \frac{2\omega}{(D+\omega)^3} + \frac{\omega^2}{(D+\omega)^4} \right\}$$
$$\leq \frac{2+2M}{D} + \frac{1+2M+M^2}{D^2}$$
$$\limsup_{D \to \infty} D \times \frac{E_G(x) - E_F(x)}{E_G(x)} \leq \limsup_{D \to \infty} 2(1+M) + \frac{M^2 + 2M + 1}{D^2} \to 2(1+M)$$

For the volume, we have:

$$\frac{V_G(x) - V_F(x)}{V_G(x)} \leq 1 - \left( 1 - \max_{v \in \mathcal{T}_x M} \left\{ \frac{1}{D+\omega} + \frac{\omega}{(D+\omega)^2} \right\} \right)^q$$
$$\leq 1 - \left( 1 - \frac{1+M}{D} \right)^q$$

Using Taylor series expansion, when $x \sim 0$, we have: $1 - (1-x)^q = qx + o(x^2)$. Let's call $\varepsilon \ll 1$, and rewrite the Taylor series:

$$\frac{V_G(x) - V_F(x)}{V_G(x)} \leq q\frac{1+M}{D} + \varepsilon q\frac{1+M}{D}$$
$$\limsup_{D \to \infty} \frac{D}{q} \times \frac{V_G(x) - V_F(x)}{V_G(x)} \leq (1+M)(1+\varepsilon)$$

The difference between the functionals can converge to zero if they are similar in high dimensions, or if they all diverge to infinity. This latter case does not happen as we assume the latent manifold being bounded, and so the metrics are then finite, which concludes the proof.

$\square$

**Corollary 3.4.** Let $f$ be a Gaussian Process. In high dimensions, the relative ratio between the Finsler norm $\|\cdot\|_F$ and the Riemmanian norm $\|\cdot\|_R$ is:
$$\frac{\|v\|_R - \|v\|_F}{\|v\|_R} = \mathcal{O}\left(\frac{1}{D}\right)$$

And, when $D$ converges toward infinity: $\forall v \in \mathcal{T}_z \mathcal{Z}$, $\|v\|_R \underset{+\infty}{\sim} \|v\|_F$.

**Proof.** Similar to the 3.3, assuming that our latent space is bounded, from B.2.2, we have $0 \leq \omega \leq MD$. From 3.3, we deduce:

$$0 \le \frac{\|v\|_R - \|v\|_F}{\|v\|_R} \le \frac{1}{D + \omega} + \frac{\omega}{(D + \omega)^2}$$
$$\le \frac{1 + M}{D}$$

In a bounded manifold, the metric are finite. We can deduce that they converge to each other in high dimensions. $\qquad\square$

## C  Experiments

### C.1  Datasets

#### C.1.1  Font data

The dataset represents 46 different font for each letter (upper and lower case) whose contour is parametrised by a spline (or two splines, depending on the letter used) obtained from at least 500 points Campbell & Kautz (2014).

In our case, we choose to learn the manifold of the letter **f**. The dataset is composed of 46 different fonts, each letter being drawn by 1024 points. We reduce this number from 1024 to 256 by sampling one point every 4. The dimension of the observational space is then 256.

#### C.1.2  qPCR

The qPCR data, gathered from Guo et al. (2010), was used to illustrate the training of a GPLVM in Pyro Pyro (2022) and is available at the Open Data Science repository Ahmed et al. (2019). It consists of 437 single-cell qPCR data for which the expression of 48 genes has been measured during 10 different cell stages. We then have 437 data points, 48 observations, and 10 classes. Before training the GP-LVM, the data is grouped by the capture time, as illustrated in the Pyro documentation.

#### C.1.3  Pinwheel on a sphere

A pinwheel in 2-dimension is created and then projected onto a sphere using a stereographic projection method. The final dataset is composed of 1000 points with their coordinates in 3-dimensions.

### C.2  GP-LVM training

We learn our two-dimensional latent space by training a GP-LVM Lawrence (2003) with Pyro Bingham et al. (2019). The Gaussian Process used is a Sparse GP, defined with a kernel (RBF, or Matern) composed of a one-dimensional lengthscale and variance. The parameters are learnt with the Adam optimiser Kingma & Ba (2014). The number of steps and the initialisation of the latent space vary with the dataset.

| datasets | pinwheel | font data | qPCR |
|---|---|---|---|
| Number of data points | 500 | 46 | 437 |
| Number of observations | 3 | 256 | 48 |
| initialisation | PCA | PCA | custom |
| kernel | RBF | Matern52 | Matern52 |
| steps | 17000 | 5000 | 5000 |
| learning rate | 1e-3 | 1e-4 | 1e-4 |
| lengthscale | 0.24 | 0.88 | 0.15 |
| variance | 0.95 | 0.30 | 0.75 |
| noise | 1e-4 | 1e-3 | 1e-3 |

Table 2: Description of the datasets trained with a GP-LVM.

### C.3 Computing indicatrices

An indicatrix of a function $g$ at a point $x$ is defined such that: $v \in \mathcal{T}_x M | g_x(v) < 1$. In other terms, the indicatrix is the representation of a unit ball in our latent space. If we use an euclidean metric, our indicatrix in our 2-dimensional latent space would be a unit ball, as we need to solve: $v \in \mathcal{T}_x M, \|v\| < 1$. For a Riemannian metric, our indicatrix is necessarily an ellipse, whose semi axis are the square-roots of the eigenvalues of the metric tensor $G$:: $v \in \mathcal{T}_x M, v^\top G v < 1$. For our Finsler metric, we don't have an analytical solution, and so it's difficult to predict the shape of the convex polygon.

In this paper, the indicatrices are drawn the following way: for a single point in our latent space, we compute the value of $v^\top G v$ and $F(x, v)$ for $v$ varying over the space. We then extract the contour when $v^\top G v$ and $F(x, v)$ are equal to 1. Computing the area of the indicatrices will be used in the section C.4 to compute the volume measures.

### C.4 Computing the volume forms

For the figures used in this paper, by default, the background of the latent space represents the volume measure of the expected Riemannian metric ($V_G = \sqrt{\mathbb{E}[G]}$) on a logarithm scale. In figure 3, the volume measure of the Finsler metric is also computed.

#### C.4.1 Finsler metric

To compute the volume measure of our Finsler metric, we choose the Busemann-Hausdorff definition, which is the ratio of a unit ball over the volume of its indicatrix: $\mathcal{V} = \text{vol}(\mathbb{B}^n(1))/\text{vol}(\{v \in \mathcal{T}_x M | F(x, v) < 1\})$. While our Finsler function has an analytical form, its expression doesn't allow to directly solve the equation: $v \in \mathcal{T}_x M, F(x, v) < 1$. Instead we approximate its indicatrix as describe in C.3, using a contour plot and extracting the paths vertices. We can then compute the area of the obtained polygon, and divide with the volume of a unit ball: $\text{vol}(\mathbb{B}^2(1)) = \pi$.

The volume measure can then be computed for each point over a grid (32 x 32, in figure 3), and we interpolate all the other points. Note that this method can only be used when our latent space is of dimension 2.

#### C.4.2 Expected Riemannian metric

There is two ways to compute the volume measure of the expected Riemannian metric. One way is to directly use the metric tensor: $V_G = \sqrt{\mathbb{E}[G]}$. Another one is to remember that any Riemannian metric is a Finsler metric, and thus, the Busemann-Hausdorff definition also applied for our metric: $V_G = \text{vol}(\mathbb{B}^n(1))/\text{vol}(\{v \in \mathcal{T}_x M | v^\top \mathbb{E}[G] v < 1\})$. Solving $v^\top \mathbb{E}[G] v < 1$ for $v \in \mathcal{T}_x M$ is equivalent to solving the area of an ellipse.

For the first method, we can either sample multiple times the metric, which is computationally expensive, or use the fact that our metric tensor is a non-central Wishart matrix: $G = J^\top J \sim \mathcal{W}_q(D, \Sigma, \Sigma^{-1}\mathbb{E}[J]^\top \mathbb{E}[J])$, with $\Sigma$ the covariance of the Jacobian $J$ and $D$ the dimension of the observational space. In this case, its expectation is: $\mathbb{E}[G] = \mathbb{E}[J]^\top \mathbb{E}[J] + D\Sigma$. We can access the derivatives of the function $f$ (detailed in section D.2), and compute both quantities $\mathbb{E}[J]$ and $\Sigma$ needed to estimate the expected metric and its determinant.

For the second method, we can compute the area of the ellipse in the same way we compute the Finsler volume measure.

### C.5 Experiments when increasing the number of dimensions

In Figure 4, we computed the volume ratio and draw indicatrices while varying the number of dimensions, to illustrate that both our Finsler metric and the expected Riemannian metric seem to converge when $D$ increases.

The main issue with this experiment is to vary only one factor, the number of dimensions $D$, while keeping the other factors unchanged. This is difficult for two reasons: 1) in real toy examples as seen in Figure 5, even with a very low dimensional observational such as in the pinwheel on the sphere, both metrics are already

very similar to each other. It would be difficult to illustrate a convergence while increasing the number of dimensions. 2) the function $f$ needs to be learnt again each time we increase the number of dimensions of the observational space, and the parameters of the Gaussian Process will change too.

Instead, we try to illustrate our results by computing empirically the stochastic metric tensor $G = J^\top J$, using its Jacobian $J \sim \mathcal{N}(\mathbb{E}[J], \Sigma)$, a $D \times q$ matrix. The number of dimensions is modified by simply truncatenating the Jacobian $J$. In Figure 4, the volume ratio is computed for 12 Jacobians obtained with different random parameters $\mathbb{E}[J]$ and $\Sigma$. The Finsler and Riemannian indicatrices (lower right) are drawn for only one Jacobian selected randomly.

## D    Computations

### D.1    Computing geodesics with Stochman and minimising the curve energy functionals

An essential task is to compute shortest paths, or geodesics, between data points in the latent space. Those shortest paths can be obtained in two ways: either by solving a corresponding system of ODEs, or by minimising the curve energy in the latent space. The former being computationally expensive, we favour the second approach which consists in optimising a parameterised spline on the manifold. This method is already implemented in Stochman Detlefsen et al. (2021), where we can easily optimise splines by redefining the curve energy function of a manifold class.

We need two curvge energy functional: one for the expected Riemannian metric and one for the Finsler metric.

### D.1.1    Curve energy for the Riemannian metric

We know that the stochastic metric tensor $G_t$ defined on a point $t$ follows a non-central Whishart distribution. Thus, we can compute its expectation $\mathbb{E}[G_t]$ knowing the Jacobian covariance and expectation: $\mathbb{E}[J_t]$ and $\Sigma$. The next Section D.2 explains how to compute those quantities.

Assuming the spline is discretized into N points, we can compute the curve energy with:

$$E_G(\gamma(t)) = \int_0^1 \dot{\gamma}(t)^\top \mathbb{E}[G_t]\dot{\gamma}(t)\, dt \approx \sum_{i=1}^N \dot{\gamma}_i^\top \left(\mathbb{E}[J_i]^T \mathbb{E}[J_i] + D\Sigma_i\right) \dot{\gamma}_i.$$

### D.1.2    Curve energy for the Finsler metric

In order to compute of the curve energy $\mathcal{E}(\gamma)$, we must first derive the expectation $\mathbb{E}[J_t]$ and covariance $\Sigma$ of the Jacobian of $f$, which should follow a normal distribution: $J_i = \partial f_i \sim \mathcal{N}\left(\mathbb{E}[J], \Sigma\right)$. We assume the points $\partial f_i$ are independent samples with the same variance drawn from a normal distribution. We can then compute the Finsler metric which follows a non-central Nakagami distribution (See Proposition 2.2):

$$\mathcal{E}(\gamma(t)) = \int_0^1 F(t, \dot{\gamma}(t))^2\, dt \approx \sum_{i=1}^N 2\dot{\gamma}_i^\top \Sigma_i \dot{\gamma}_i \left(\frac{\Gamma(\frac{D}{2} + \frac{1}{2})}{\Gamma(\frac{D}{2})}\right)^2 {}_1F_1\left(-\frac{1}{2}, \frac{D}{2}, -\frac{\omega_i}{2}\right)^2,$$

with ${}_1F_1$ the confluent hypergeometric function of the first kind and $\omega_i = (\dot{\gamma}_i^\top \Sigma_i \dot{\gamma}_i)^{-1}(\dot{\gamma}_i^\top \Omega_i \dot{\gamma}_i)$ and $\Omega_i = \Sigma_i^{-1}\mathbb{E}[J_i]^\top \mathbb{E}[J_i]$.

This function has been implemented in Pytorch using the known gradients for the hypergeometric function: $\frac{\partial}{\partial x}{}_1F_1(a, b, x) = \frac{a}{b}{}_1F_1(a+1, b+1, x)$.

### D.2    Accessing the posterior derivatives

We assume that the probabilitic mapping $f$ from the latent variables $X$ to the observational variables $Y$ follows a normal distribution. We would like to obtain the posterior kernel $\Sigma_*$ and expectation $\mu_*$ such that $p(\partial_t f | Y, X) \sim \mathcal{N}(\mu_*, \Sigma_*)$.

We make the hypothesis the observed variables are modelled with a gaussian noise $\epsilon$ whose variance is the same in every dimension. In particular, for the $n^{th}$ latent $(x)$ and observed $(y)$ variable in the $j^{th}$ dimension: $y_{n,j} = f_j(x_{n,:}) + \epsilon_n$. Thus, the output variables have the same variance, and the posterior kernel $\Sigma_*$ is then isotropic with respect to the output dimensions: $\Sigma_* = \sigma_*^2 \cdot I_D$.

There are two ways of obtaining the posterior variance and expectation:

- We use the gaussian processes to predict the derivative $(\partial_c f)$ of the mapping function $f$, and we multiply the obtained posterior kernel by the curve derivative $(\partial_t c)$, following the chain rule: $\frac{df(c(t))}{dt} = \frac{df}{dc} \cdot \frac{dc}{dt}$ (Section: D.2.1)

- We discretize the derivative of the mapping function as the difference of this function evaluated at two close points. We use a linear operation to obtain the posterior variance and expectation: $\partial_t f(c(t)) \sim f(c(t_{i+1})) - f(c(t_i))$. (Section: D.2.2)

### D.2.1    Closed-form expressions

We assume that $f$ is a Gaussian process. Hence, because the differentiation is a linear operation, the derivative of a Gaussian Process is also a Gaussian Process Rasmussen & Williams (2005).

The data $Y \in \mathcal{R}^{N \times D}$ follows a normal distribution, so we can infer the partial derivative of one data point $(J^T)_{ji} = \frac{\partial y_i}{\partial x_j}$, with $i = 1 \ldots D$ and $j = 1 \ldots d$. We have:

$$
\left[ \begin{array}{c} Y \\ (J)^\top \end{array} \right] = \prod_{i=1}^{D} \mathcal{N} \left( \left[ \begin{array}{c} \mu_y \\ \mu_{\partial y} \end{array} \right], \left[ \begin{array}{cc} K(x,x) & \partial K(x,x_*) \\ \partial K(x_*,x) & \partial^2 K(x_*,x_*) \end{array} \right] \right).
$$

and $J^\top$ can be predicted:

$$
p(J^\top | Y, X) = \prod_{i=1}^{D} \mathcal{N} \left( \mu_*, \Sigma_* \right),
$$

with:

$$
\mu_* = \partial K(x_*,x) \cdot K(x,x)^{-1} \cdot (y - \mu_y) + \mu_{\partial y}
$$
$$
\Sigma_* = \partial^2 K(x_*,x_*) - \partial K(x_*,x) \cdot K(x,x)^{-1} \cdot \partial K(x,x_*).
$$

Finally, $\partial_t f$ is obtained:

$$
p(\partial_t f(c(t)) | f(x), x) = \prod_{i=1}^{D} \mathcal{N} \left( \dot{c} \mu_*, \dot{c}^\top \Sigma_* \dot{c} \cdot I_D \right).
$$

### D.2.2    Discretization

One can notice that: $\partial_t f(c(t)) \sim f(c(t_{i+1})) - f(c(t_i))$. We know that $f(c(t_{i+1}))$ and $f(c(t_i))$ both follows a normal distribution.

$$
\left[ \begin{array}{c} f(c(t_i)) \\ f(c(t_{i+1})) \end{array} \right] = \prod_{j=1}^{D} \mathcal{N} \left( \left[ \begin{array}{c} \mu_i \\ \mu_{i+1} \end{array} \right], \left[ \begin{array}{cc} \sigma_{ii}^2 & \sigma_{i,i+1}^2 \\ \sigma_{i+1,i}^2 & \sigma_{i+1,i+1}^2 \end{array} \right] \right).
$$

If $Y = AX$ affine transformation of a multivariate Gaussian $X \sim \mathcal{N}(\mu, \sigma^2)$, then Y is also a multivariate Gaussian with: $Y \sim \mathcal{N}(A\mu, A^T \sigma^2 A)$. In our case, we choose $A^T = [-1, 1]$. We have:

$$
f(c(t_{i+1})) - f(c(t_i)) \sim \mathcal{N}(\mu_*, \sigma_*^2 \cdot I_D),
$$

with:

$$\mu_* = \mu_{i+1} - \mu i$$
$$\sigma_*^2 = \sigma_{ii}^2 + \sigma_{i+1,i+1}^2 - 2\sigma_{i,i+1}^2$$

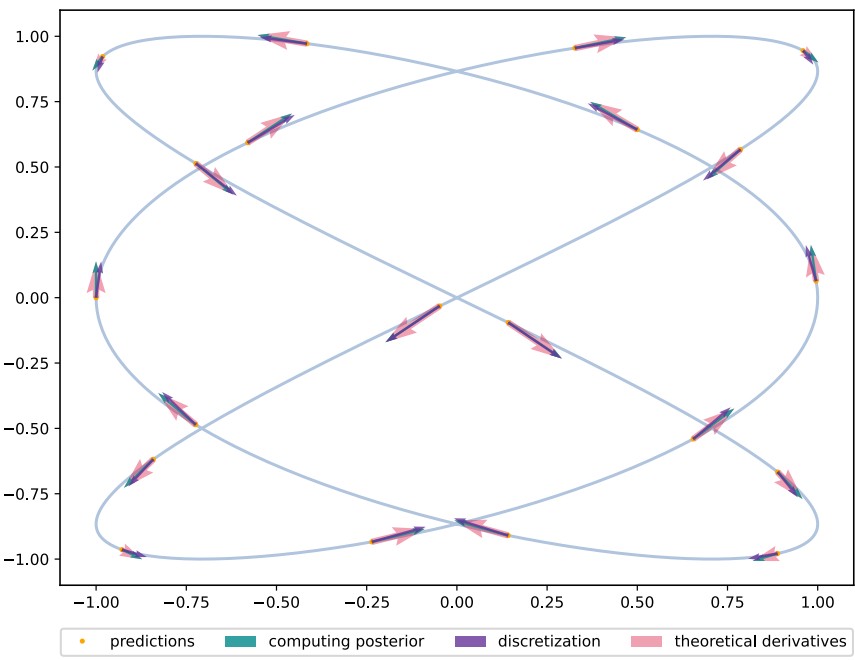

Figure 11: Illustration of the derivatives obtained with a trained GP on a simple parametrised function: both methods give the correct derivatives if enough points are sampled.

