# OpenReview forum: "Identifying latent distances with Finslerian geometry"
_TMLR — Rejected by TMLR_

### Review · Reviewer_GAcf · 2023-02-24

**Summary Of Contributions:**

This paper studies the geometry of latent space common to probabilistic models where the decoding map is stochastic. The paper proposes a new metric, defined via the expected curve length. This leads to a Finsler geometry on the latent space. The paper then derives the bounds on the Finsler metric, curve lengths, energy and volume measures, with respect to a pulled back Riemannian metric. The paper performs various experiments illustrating the bounds and comparing the Finsler metric with the Riemannian metric through GPLVM.

**Audience:**

No

**Claims And Evidence:**

Yes

**Requested Changes:**

Majors:

(1) Include experiments showing the difference between Finsler and Riemannian metric, for example by plotting the geodesics in the high-variance areas.

(2) Include experiments demonstrating the benefits of using Finsler metric instead of the Riemannian metric. It would be better to also add Euclidean metric and comment on the utility among the metrics.

Minors:

(a) What does Figure 2 mean? The left represents expected Riemannian metric and the right represents expected curve lengths, which are not comparable. Would it be possible to plot the curve length from expected Riemannian metric versus expected curve length?

(b) In proof of Proposition 3.1, F(x,v) → F_x v.

(c) In caption of Figure 4, there are two “(A)”.

(d) In the first line of Section 4, “For thi”.

(e) In Section 5, “Riemanniangeometry”.



**Strengths And Weaknesses:**

Strengths:

(1) The proposed Finsler metric is new and interesting, and appears naturally derived from the expected curve length.

(2) The comparisons between the Finsler metric and the pullback Riemannian metric are clearly derived.

(3) It is interesting to see results such as Proposition 3.2 and 3.3 showing when two metrics become identical.

Weaknesses:

(1) It is not clear to me the motivation of considering the proposed metric, compared to the Riemannian metric analysed by Eklund & Hauberg (2019). This is even worse when the computation of Finsler metric is more difficult than the Riemannian metric, as also noticed by the authors. Hence it is unclear whether the work in its current form is interesting for some audiences.

(2) The main goal of Eklund & Hauberg (2019) is to equip the latent space with a geometric structure, such that one can interpolate and measure the distance between the latent representations. It appears, at least from the current version of the paper, that the new metric does not add much value in this respect. This is particularly noticed in the real experiments where the geodesics between the two geometries match exactly (in regions of low variance).

---

> ### Author Response · Authors · 2023-03-07
> **Reply to reviewer GAcf**
>
> Thank you a lot for your feedback and valuable comments!
>
> `` (1) It is not clear to me the motivation of considering the proposed metric, compared to the Riemannian metric analysed by Eklund & Hauberg (2019). This is even worse when the computation of Finsler metric is more difficult than the Riemannian metric, as also noticed by the authors. Hence it is unclear whether the work in its current form is interesting for some audiences. ``
> Thank you a lot for raising this concern. We modified the introduction and section 2 to make it clearer.  Eklund & Hauberg (2019) only compared the latent distances (curve lengths) obtained through the stochastic pullback metric, and they show that the difference between those distances is upper bounded by the number of dimensions, which is recalled in Proposition 2.1.
> We are extending this work in several ways: (1) we are not only comparing the latent distances, but the norms those functionals are derived from, (2) we show that the expected norm is a very special metric, which is a Finsler metric, (3) for $f$ a Gaussian process, we show that the expected norm has a closed-form expression, and (4) we show that, for $f$ a Gaussian process, the difference between the Finsler and the Riemannian norms are upper bounded by the number of dimensions, and so are all the derived functionals.
>
> `` (2) The main goal of Eklund & Hauberg (2019) is to equip the latent space with a geometric structure, such that one can interpolate and measure the distance between the latent representations. It appears, at least from the current version of the paper, that the new metric does not add much value in this respect. `` Our proposed metric is actually the norm that leads to the derivation of the curve-length, studied by Eklund and Hauberg (2019). Yet, they didn't make the link between Finsler geometry and their work.
> ``This is particularly noticed in the real experiments where the geodesics between the two geometries match exactly (in regions of low variance). `` Yes, it is true that both geometries seem to match in practice even when the data space has few dimensions, and we believe it is an interesting result.
>
> ``(1) Include experiments showing the difference between Finsler and Riemannian metric, for example by plotting the geodesics in the high-variance areas.``. This is an excellent idea, and we will aim for more experiments by the end of the rebuttal period. We would love to have an extension from the Area Chair if possible.
>
> `` (2) Include experiments demonstrating the benefits of using Finsler metric instead of the Riemannian metric. It would be better to also add Euclidean metric and comment on the utility among the metrics.`` It is, again, an excellent suggestion, and we will aim for those experiments. As a side comment, we can see that the Riemannian energy of a curve $\gamma$ is equal to the Finsler energy plus a variance term: $E_R = E_F + \int \text{Var}[|| \dot{\gamma_t}||_G] dt$. When we want to obtain a geodesic, in practice, we tend to minimise the curve-energy. Here, we can see that the Riemannian curve energy is penalised by the variance: the Riemannian curve will want to avoid area of uncertainty. One potential application that could highlight the benefits of the Finsler metric might be active learning, where we aim to fetch data points with high uncertainty to have a richer model.
>
> `` (a) What does Figure 2 mean? The left represents expected Riemannian metric and the right represents expected curve lengths, which are not comparable. Would it be possible to plot the curve length from expected Riemannian metric versus expected curve length? ``. It is a good remark, and this illustration might indeed add more confusion than clarity, so we decided to remove it.
>
> `` (b) In proof of Proposition 3.1, F(x,v) → F_x v. `` Thank you for spotting this ! Many notations have changed in order to make the paper clearer, and notably, the Finsler norm is $F(\cdot)$, and the norm induced by the expected Riemannian metric tensor $\sqrt{g(\cdot, \cdot)}$ are now noted: $||\cdot||_F$ and $||\cdot||_R$.
>
> `` (c) In caption of Figure 4, there are two “(A)” ``. It is corrected.
>
> `` (d) In the first line of Section 4, “For thi”. ``. It is corrected.
>
> `` (e) In Section 5, “Riemanniangeometry”. ``. It is corrected, thank you a lot !

---

### Review · Reviewer_k87i · 2023-02-25

**Summary Of Contributions:**

This paper suggests using a Finsler Metric instead a Riemannian metric. Specifically, the paper suggests that when we have a stochastic map $f$ from a latent space manifold to observed data, then if we look at the pullback metric of this map then due to the stochasticity of the map $f$, we have a random variable as the metric. The paper says that we could look at the Riemannian approximation given by the expected metric or use the Finsler metric instead. The paper then shows that the two metrics result in similar geodesics in data-dense areas and prove upper bounds on the error. However, the two differ in sparse data regions.

**Audience:**

Yes

**Claims And Evidence:**

Yes

**Requested Changes:**

Please see weaknesses above.

**Strengths And Weaknesses:**

**Strengths**
---

The paper presents bounds on the difference between the two different metrics. Hence, in my opinion, showing that in data-dense regions, using either metric is okay.

I think the paper provides a good set of tools someone could use to build on.

I appreciate the proof sketches in the main text.

**Weaknesses**
---

While mainly easy to follow, I would say that the writing of the paper is still a weakness. Here I detail some of my concerns.

1) In the introduction, the paper has a sentence in bold "we argue that doing so using the Euclidean metric in the latent space
is wrong and misleading. " However, the paper isn't really about this. That is, the paper is about Riemannian vs. Finsler and not Riemannian vs. Euclidean. The only place this argument shows up is in Figure 1, *which is not referenced in the main text*. Hence my first suggestion is to rephrase this bit. Specifically, I think it is reasonably well accepted now that using the appropriate metric is important and that we cannot just use the Euclidean metric.

2) Figures: I think the figures are great, but they should be appropriately placed and referenced in the main text. Specifically, Figures 1 and 2 are not discussed in the main text, and I think such a discussion could help provide context. Second, Figure 4 is referenced before Figure 3, and Figure 5 appears much after it is referenced (There are also typos in the caption for Figure 5.)

3) Background and definitions. There are a few terms that should be defined. Specifically, I think section A.3 should be moved to the main text. This will help readers realize that the definitions in 2.5 correspond to the version for the Finsler metric that appears *later*. The paper assumes we have made this connection, but it would be good to state this explicitly.

Further things include other terms that should be defined, such as non-central Wishart (which I assume is a wishers matrix $XX^T$ in which the entries of $X$ do not have mean zero), Gaussian process, and hypergeometric function of the first kind. These should have references for the definition or simple descriptions in the text. I think this will help expand the audience the paper can reach.

**Questions**
---

1) The pullback metric depends on the metric in the target manifold. If I understood correctly throughout this paper, the assumption is that the target metric is the standard Euclidean inner product. Is that correct? Could the authors comment on what, if anything, changes if the target metric is something else?

2) The $\omega$ term in Proposition 2.3, 3.3, and Corollary 3.2 is seemingly treated as a constant but depends on both the point on the manifold $x$ and the point in the tangent space $v$. Hence I would like more clarity on this term.

3) The paper mentions that the two metrics differ in data-sparse regions. Do the authors have any potential applications or areas where they think the Finsler one would provide value?

4) Finally, as $\mathcal{D} \to \infty$, the two metrics should coincide, is this probably due to some universality result? Specifically, I assume this is the ``variance'' of $f$ is bounded as $\mathcal{D} \to \infty$. So we have some concentration of measure happening?

---

> ### Author Response · Authors · 2023-03-07
> **Reply to reviewer k87i (1/2)**
>
> Thank you for your feedback and the very interesting questions!
>
> `` In the introduction, the paper has a sentence in bold "we argue that doing so using the Euclidean metric in the latent space is wrong and misleading. " However, the paper isn't really about this. That is, the paper is about Riemannian vs. Finsler and not Riemannian vs. Euclidean. The only place this argument shows up is in Figure 1, which is not referenced in the main text. Hence my first suggestion is to rephrase this bit. Specifically, I think it is reasonably well accepted now that using the appropriate metric is important and that we cannot just use the Euclidean metric. `` We rephrased the introduction to better reference the Figure and also remove the argument about the Euclidean metric.
>
> `` Figures: I think the figures are great, but they should be appropriately placed and referenced in the main text. Specifically, Figures 1 and 2 are not discussed in the main text, and I think such a discussion could help provide context. Second, Figure 4 is referenced before Figure 3, and Figure 5 appears much after it is referenced (There are also typos in the caption for Figure 5.) `` Figure 1 is now better referenced in the introduction, and Figure 2 has been removed since it might bring some confusion (also noted from reviewer GAcf). the caption of Figure 5 is now fixed. We are afraid it might be confusing to place Figure 4 before Figure 3, as we need to first understand the indicatrices before comparing the volume two metrics in practice. An alternative would be to completely remove the reference while introducing the Busemann-Hausdorff volume. Figure 5 combines also many results to illustrate the behaviour of the metrics in high-dimensions, moving the Figure before the "high dimensional" section might be confusing too.
>
> `` Background and definitions. There are a few terms that should be defined. Specifically, I think section A.3 should be moved to the main text. This will help readers realize that the definitions in 2.5 correspond to the version for the Finsler metric that appears later. The paper assumes we have made this connection, but it would be good to state this explicitly. `` More definitions, including the one for a Finsler metric, have been added to the paper. In general, the introduction and section 2 have been rewritten to improve the clarity of the paper, and the notations have also been simplified for the Finsler norm and the norm induced by the Riemannian metric.
>
> `` Further things include other terms that should be defined, such as non-central Wishart (which I assume is a wishers matrix $XX^{\top}$
>  in which the entries of  $X$ do not have mean zero), Gaussian process, and hypergeometric function of the first kind. These should have references for the definition or simple descriptions in the text. I think this will help expand the audience the paper can reach. `` Those terms are now defined, and, indeed, it improves the clarity of the paper: thank you for this suggestion!

---

> ### Author Response · Authors · 2023-03-07
> **reply to reviewer k87i (2/2)**
>
>
> `` The pullback metric depends on the metric in the target manifold. If I understood correctly throughout this paper, the assumption is that the target metric is the standard Euclidean inner product. Is that correct? Could the authors comment on what, if anything, changes if the target metric is something else? `` You are absolutely correct, and it is a very good question! The target metric is indeed Euclidean, and the stochastic pullback metric tensor is defined this way: $G_1 = J^{\top} \cdot I_D \cdot J$, with $J$ the jacobian of our map $f$ and $I_D$, the identity matrix. Now, let's assume we have another Riemannian target metric called $M$. The pullback metric would now be defined as $G_2 = J^{\top} \cdot M \cdot J$. In both cases, $G_1$ and $G_2$ are both random Riemannian metrics and, when $f$ is a Gaussian process, they are both non-central Wishart. Interestingly, all the results hold, as long as the target metric is Riemannian (to ensure that the pullback metric is Riemannian too). Thank you for this question!
>
> `` The $\omega$ term in Proposition 2.3, 3.3, and Corollary 3.2 is seemingly treated as a constant but depends on both the point on the manifold $x$ and the point in the tangent space $v$. Hence I would like more clarity on this term. ``  It is correct that $\omega$ vary depending on many parameters. The paper has been updated to give more intuition about this term, just after Proposition 2.3., when introducing the hypergeometric function $_1F_1$. $\omega = (v^{\top}\Sigma v)^{-1} (v^{\top} \mathbf{E}[J]^{\top}\mathbf{E}[J]v)$ appears from the non-central Wishart distribution, and when the distribution (of the Jacobian) shift away from the origin, it represents the magnitude and the direction of this shift, balanced by the correlation between the data. I am afraid I can't give more intuition about this term. Yet, when making the assumption that the manifold is bounded, we know (Lemma B.2.2) that $\omega$ doesn't grow faster than the number of dimensions, and this helps us to prove Corollary 3.3. and 3.4.
>
> `` Finally, as $D \to \infty$, the two metrics should coincide, is this probably due to some universality result? Specifically, I assume this is the "variance" of $f$ is bounded as $D \to \infty$. So we have some concentration of measure happening? ``. Yes! We also believe this is exactly a specific case of concentration of measure, but we were unable to prove it so far. We are wondering if, in general, a non-central Wishart matrix $G = X_{ji}X_{ij}$ when $i\in[1,D]$ and $D\to\infty$ becomes deterministic. The central limit theorem seems to be easily applicable for the diagonal terms of the random matrix, but not necessarily for the non-diagonal ones. If it is a specific case of concentration of measure, it would explain why both metric converge, since $\mathbf{E}[G] \approx G$, and we would love to push in that direction, in the future.

---

> > ### Comment · Reviewer_k87i · 2023-03-10
> > **Thank you for the answer**
> >
> > Thanks

---

### Review · Reviewer_vUJf · 2023-02-26

**Summary Of Contributions:**

This paper analyzes the geometry of stochastic generative models. In particular, it defines a Finslerian Metric on the immersion induced by the latent variable model f and analyzes various properties of the metric.

**Audience:**

Yes

**Claims And Evidence:**

No

**Requested Changes:**

In addition to the above weaknesses, which should be addressed, there are some typos:

* Figure 4: the (A) is used instead of (C) and (D)


**Strengths And Weaknesses:**

Strengths
-----------
* The proposed construction is novel.
* The writing is reasonable clear and concise.


Weaknesses
--------------
* The paper doesn't properly introduce the random Riemannian metric. While references are given to previous work, it would be helpful to elucidate how VAEs or GPs induce this metric.
* Related to the above point, the paper also introduces the concept of f as both an immersion and a stochastic process rather suddenly. In particular, I find it hard to parse this construction since a stochastic process is time-indexed. The definition in the appendix is also not very helpful. What is the expected value taken over? Time?
* Several theoretical results from the paper (e.g. the O(1/D) convergence result) seem to be heavily reliant on the gaussian process metric of Tosi et al. 2014. While this is mostly fine, several introductions give the impression that the results are more general. Furthermore, it should be noted that most of the results seem to break under more general immersions.
* I don't think the lower bound of Proposition 3.1 is correct. In particular, where does the 1F1 hypergeometric function from Proposition 2.3 go?
* Corollaries 3.3 and 3.4 obviate several critical details. In particular, the value of $\omega$ in Prop 3.3 seems to change the convergence depending on how it relates to $D$. Notably, one could construct an arbitrarily bad Jacobian matrix for each dimension for which the error shouldn't converge.
* The experiments seem rather toy. Something more real-world could show more relevancy.
* The experiments don't show show how this is an improvement over the previous Riemannian metric.

Overall Thoughts
--------
The paper proposes an alternative to the Riemannian metric defined for stochastic manifolds. However, the paper mostly shows that the proposed approach is similar to the predefined value (under some circumstances), without showing if this formulation is useful.

---

> ### Author Response · Authors · 2023-03-07
> **Reply to reviewer vUJf**
>
> Thank you a lot for your feedback!
>
> `` The paper doesn't properly introduce the random Riemannian metric. While references are given to previous work, it would be helpful to elucidate how VAEs or GPs induce this metric. Related to the above point, the paper also introduces the concept of f as both an immersion and a stochastic process rather suddenly. In particular, I find it hard to parse this construction since a stochastic process is time-indexed. The definition in the appendix is also not very helpful. What is the expected value taken over? Time?`` Thank you for your comment. We modified Section 2, such that a _stochastic process_ and _random Riemannian metric_ are better explained and introduced.
>
> `` Several theoretical results from the paper (e.g. the O(1/D) convergence result) seem to be heavily reliant on the gaussian process metric of Tosi et al. 2014. While this is mostly fine, several introductions give the impression that the results are more general. Furthermore, it should be noted that most of the results seem to break under more general immersions. `` This is a very good point. For each theorem, we now re-introduce $f$, defined either as a stochastic process or as a Gaussian process, to avoid any confusion.
>
> `` I don't think the lower bound of Proposition 3.1 is correct. In particular, where does the 1F1 hypergeometric function from Proposition 2.3 go? ``. Thank you so much for your careful review! There is actually a typo in the proof: the Pochhammer symbols are defined as rising factorials, and so: $\frac{(a)_k}{(b)_k} = \frac{\Gamma(a+k)}{\Gamma(b+k)}\frac{\Gamma(b)}{\Gamma(a)}$ ($\Gamma(a)$ and $\Gamma(b)$ were inverted), and so, the $\Gamma$-functions now cancel each other in the next equations. If the proof is still not clear, we can add more details.
>
> ``Corollaries 3.3 and 3.4 obviate several critical details. In particular, the value of $\omega$ in Prop 3.3 seems to change the convergence depending on how it relates to $D$. Notably, one could construct an arbitrarily bad Jacobian matrix for each dimension for which the error shouldn't converge.`` It is very true that $\omega$ changes with the number of dimensions, yet the results hold because $\omega$ does not grow faster than the dimensions: $\omega \leq m \cdot D$, with $m\in\mathbf{R}_+$ an arbitrary constant (See Lemma B.2.2.). This comes from the fact that we assume our latent manifold to be bounded, which means that any element of Jacobian is upper bounded by any arbitrary constant.
>
> `` The experiments seem rather toy. Something more real-world could show more relevancy. `` If the TMLR Action Editor agrees for an extension, we would love to improve our paper with more experiments.
>
> `` The experiments don't show show how this is an improvement over the previous Riemannian metric. `` The experiments indeed highlight the fact that both metrics are very similar, even when the data is low-dimensional. Additionally, we don't necessarily think that one metric is better than the other, but simply that it seemed more sensible to take the expectation of the norm instead of taking the expectation of the metric tensor, who serves as a proxy to define the Riemmanian metric.
>
> `` Figure 4: the (A) is used instead of (C) and (D) ``. Thank you, it is now corrected.

---

### Review · Reviewer_5EPE · 2023-02-28

**Summary Of Contributions:**

This paper defines latent space geometry in deep generative models and related models. When the mapping from latent space to the observables is random, the induced geometry is also random. One can either define the geometry by taking the expectation of the induced random metric tensor (previous work), or by taking the expectation on the local norm on the tangent space (this paper). The authors discussed basic properties of the induced norm (Finsler metric) through the GPLVM model (most are straightforward) with intuitive demonstrations of the latent space defined.

**Audience:**

Yes

**Broader Impact Concerns:**

This paper is of theoretical nature with limited ethical impact.

**Claims And Evidence:**

Yes

**Requested Changes:**

Introduction: explain why defining a good latent geometry is important and how it can be used in deep learning practice.

The notion of "immersion" needs some explanation in the deep learning context.

Related work is unfortunately too brief (2 papers in Finsler geometry in machine learning and 2 papers in defining latent space geometry). I suggest the authors either try to extend it, or try to argue that what is done here is completely novel.

The notion of "tangent space" needs an explanation (I understand the authors are taking the informal approach but still as it was not introduced before).

End of 2.2, briefly mention the meaning of the three functionals as they correspond to the deterministic case.

Throughout the paper, ${\sqrt{}}^2$ looks unnatural. Can it be simplified?

Proposition 2.1. Needs some explanation of the meaning of the statement.

Proposition 2.3, needs the explicit formula of $f$ (or its Jacobian).

Definition 3.1: introduce the notation $vol()$

Proposition 3.3 same as above for $f$. jacobian -> Jacobian

Experiments: How one can observe the Finsler geometry is better than the expected Riemannian geometry?

Throughout the paper: some places should use \citep instead of \cite to make the text coherent.

**Strengths And Weaknesses:**

Pros:
-New treatment of latent space geometry.
-Intuitive visualizations and examples.

Con:
-Self-contentedness can be improved in the main paper (many loopholes currently!)

---

> ### Author Response · Authors · 2023-03-07
> **Reply to reviewer 5EPE**
>
> Thank you a lot for your feedback and for taking the time to read the paper !
>
> `` Introduction: explain why defining a good latent geometry is important and how it can be used in deep learning practice. ``: This is intuitively explained on Figure 1, on the left panel. The introduction has been modified to better reference this Figure. Otherwise, the concept of latent geometry in machine learning is well explained by (Hauberg, 2018), referenced in the introduction. It is also well explained for VAEs by (Arvanitidis et al, 2018). I am afraid a more in-depth explanation might make the paper too dense and less readable.
>
> `` The notion of "immersion" needs some explanation in the deep learning context. ``: The term immersion is now better introduced and defined at the beginning of section 2.1. We need $f$ to be an immersion in order to properly define a Riemannian metric tensor, which needs to be positive definite.
>
> `` Related work is unfortunately too brief (2 papers in Finsler geometry in machine learning and 2 papers in defining latent space geometry). I suggest the authors either try to extend it, or try to argue that what is done here is completely novel. ``: The papers about Finsler geometry in Machine Learning are quite distinct from our work. They don't consider the stochasticity of the data, which is the main core of our paper, they are either application-focused (Ratliff et al, 2021), or they consider graph-embeddings (Lopez et al, 2021). We feel that we don't need to better introduce them. The other ones that try to deal with the stochastic geometry are only using Riemannian metrics, and we compare our work with the expected metric tensor defined by (Arvanitidis et al, 2018) and (Tosi et al, 2014). However, while they introduce this expected metric tensor, they don't study it with respect to other metrics or in high dimensions. They only use it as a way to efficiently explore their own latent space.
>
> `` The notion of "tangent space" needs an explanation (I understand the authors are taking the informal approach but still as it was not introduced before). `` The definition 2.1. now includes a short definition of a tangent plane.
>
> `` End of 2.2, briefly mention the meaning of the three functionals as they correspond to the deterministic case. `` A paragraph now includes a short intriduction to the curve length, curve energy and volume measure, just after definition 2.1.
>
> `` Throughout the paper, $\sqrt$ looks unnatural. Can it be simplified? ``. It is a very good point. One of the challenge of the paper was to find coherent notations while merging two fields, whose central mathematical object, a metric, is either define as an inner product or as a norm. In the paper, we have now replaced accordingly the Finsler metric by the norm $||u||_F = F(u)$ and the norm induced by the Riemannian inner product:  $||u||_R = \sqrt{g(u,u)}$. We hope this would clarify a lot the results and the comprehension of the paper. Thank you a lot for this valuable feedback !
>
> `` Proposition 2.1. Needs some explanation of the meaning of the statement. `` I am not sure how I can formulate this statement in a better way. (Eklund and Hauberg, 2019) worked on curve lengths, and they noted that those distances can be upper bounded by the number of dimensions.
>
> `` Proposition 2.3, needs the explicit formula of $f$ (or its Jacobian). ``: The Jacobian is now introduced as following a Gaussian distribution: $J \sim \mathcal{N}(\mathbf{E}[J], \Sigma)$. Thank you for pointing this out.
>
> `` Definition 3.1: introduce the notation $vol()$ ``. $vol$ defines the standard Euclidean volume, and is now introduced.
>
> `` Proposition 3.3 same as above for $f$. jacobian -> Jacobian``. Now corrected, thank you !
>
> `` Experiments: How one can observe the Finsler geometry is better than the expected Riemannian geometry? `` Here, the experiments actually show that both norms are very similar, even when the data space is low dimensional. We also don't argue that the Finsler norm is better than the Riemannian metric, and the introduction has been changed to better reflect this. We only thought that the Riemannian metric tensor is mostly used as a proxy to define a norm on the latent space. The concept of norm in general, either induced by the Riemannian metric or defined as a Finsler metric, is actually key to derive other geometrical quantities, and in particular, distances. Instead of taking the expectation of the metric tensor, we thought it would be more sensible to take the expectation of the stochastic norm directly.
>
> `` Throughout the paper: some places should use \citep instead of \cite to make the text coherent. `` Some citations have been changed, thank you !

---

### Author Response · Authors · 2023-03-07
**Paper updated for more clarity**

We would like to thank all the reviewers for their fast and very valuable feedback on the paper.

One important weakness is a lack of clarity. The paper could indeed be clearer if some mathematical definitions were introduced in the main text. In particular, the notions of _immersion_ and _stochastic process_ are not well understood. _Random Riemannian metric_, _Finsler metric_, and the use of _functionals_ could be defined in the main text instead of the appendix. _Non-central Wishart_, _Gaussian Process_, _Hyperbolic confluent function of the first kind_ should be quickly defined too. Those terms are now better introduced. In particular, the introduction and the section 2 have been rewritten in order to be less confusing and clearer.

In addition, the notations were confusing: in Riemannian geometry, the metric is traditionally defined as an inner product, while in Finsler geometry, we are working with a norm. Comparing those two objects seems tricky as we need to add a square root on the Riemannian metric in every proposition. We have changed the notations, and introduced the Finsler and Riemannian norms, so it seems more natural to compare those two mathematical objects.

---

> ### Author Response · Authors · 2023-03-27
> **Additional experiments shown in supplementary materials**
>
> We would like to thank the reviewers for their patience. Additional experiments have been done to address two major concerns: (1) highlighting the difference between the Finslerian and the Riemannian geodesics within area of high variance, and (2) training a GPLVM on 'more real' data (we used MNIST and Fashion MNIST).

---

### Decision · Action_Editors · 2023-04-07

**Recommendation:** Reject

**Comment:**

Ideally, I would have liked to go for the major revision option (as there is partial merit / evidence). Since it is not there in the TMLR set options, I have decided on the reject option.

I would encourage the authors to address the comments on the lack of evidence and submit the revised version. Please have a look at https://jmlr.org/tmlr/editorial-policies.html.

**Audience:**

Yes.

**Claims And Evidence:**

The paper proposes a Finsler geometry for studying stochastic modeling problems. The difference with the Riemannian approach is instead of taking the expectation of J'J, the paper studies the expectation of the norm directly (which has the Finsler interpretation). The paper subsequently compares them both. The Riemannian and Finsler norms converge in higher dimensions. The empirical results show the machinery at work.

The primary contribution of the paper has been on the theoretical study of the proposed Finsler geometry. To that end, the paper has clear evidence. All the reviewers readily agree on this. However, the evidence on why the proposed should be preferred over the Riemannian empirically is not there, and herein, lies the source of confusion on the merits. That seems to be the main concern of all the reviewers. The current revision has a toy data example, where some distinctions are brought out (but the implication still remains inconclusive). Hence, the paper in its current state does not satisfy the TMLR requirements of "accurate, convincing, and clear evidence".

---

> ### Author Response · Authors · 2023-04-17
> **Reply to the decision**
>
> Thank you for the paper handling and the support during the discussion phase.
>
> Unsurprisingly, we regret the decision, but understand that these decisions are rarely straight-forward. We, however, lack some context to better address the raised concern, and hope you can help us. Specifically, the point of our paper is to investigate a theoretical concern with the expected Riemannian metric, and we develop a Finslerian metric that avoids this theoretical issue. We then further show that even if the Finsler metric is theoretically more meaningful, it is highly similar to the expected Riemannian metric in high dimensions. We then empirically demonstrate the strong links between the two metrics, thereby validating the theoretical analysis. In this context, we struggle to understand the critique that
> > the evidence on why the proposed should be preferred over the Riemannian empirically is not there.
>
> We do not aim to make such a claim. In many ways we argue the opposite: while the Finslerian perspective solves a theoretical concern, the expected Riemannian metric is sufficiently similar that it is to be preferred in practice as it is much simpler.
>
> Given the above, it is not clear to us how we should revise the paper as we do not intend to make the statement quoted above.

---

> > ### Comment · Action_Editors · 2023-04-18
> > **Response**
> >
> > Thank you for reaching out.
> >
> > When I wrote the sentence, I had the following in mind.
> >
> > I understand that "the aim of this paper is to compare those two norms". This is mentioned on Page 2 (last paragraph). However, it is not clear in the paper whether the Finsler geometry has any benefit (especially empirical) over the Riemannian geometry, why should one care about it, or what is the point of this comparison when we have a clear choice of sticking with the Riemannian metric in practical scenarios.
> >
> > In a comparative study, ideally, we should discuss different settings where one norm should be chosen over the other or anything that compares them effectively. Now in the paper, it is mentioned that in low dimensions they are different, but what is the implication of this? Why is this interesting? These are some of the questions that are not answered clearly in the paper and the empirical evidence is missing.